# Dynamics of mTORC1 activation in response to amino acids

Maria Manifava[1†], Matthew Smith[1†], Sergio Rotondo[1], Simon Walker[1], Izabella Niewczas[1], Roberto Zoncu[2], Jonathan Clark[1], Nicholas T Ktistakis[1*]

[1]Signalling Programme, Babraham Institute, Cambridge, United Kingdom; [2]Department of Molecular and Cell Biology, University of California, Berkeley, Berkeley, United States

**Abstract** Amino acids are essential activators of mTORC1 via a complex containing RAG GTPases, RAGULATOR and the vacuolar ATPase. Sensing of amino acids causes translocation of mTORC1 to lysosomes, an obligate step for activation. To examine the spatial and temporal dynamics of this translocation, we used live imaging of the mTORC1 component RAPTOR and a cell permeant fluorescent analogue of di-leucine methyl ester. Translocation to lysosomes is a transient event, occurring within 2 min of aa addition and peaking within 5 min. It is temporally coupled with fluorescent leucine appearance in lysosomes and is sustained in comparison to aa stimulation. Sestrin2 and the vacuolar ATPase are negative and positive regulators of mTORC1 activity in our experimental system. Of note, phosphorylation of canonical mTORC1 targets is delayed compared to lysosomal translocation suggesting a dynamic and transient passage of mTORC1 from the lysosomal surface before targetting its substrates elsewhere.

**\*For correspondence:** nicholas.
ktistakis@babraham.ac.uk

[†]These authors contributed equally to this work

**Competing interests:** The authors declare that no competing interests exist.

## Introduction

Mammalian cells maintain elaborate ways to respond to amino acid availability and a prominent sensor is the protein kinase mammalian (or mechanistic) target of rapamycin complex 1 (mTORC1) (*Wullschleger et al., 2006*; *Laplante and Sabatini, 2009*). Under plentiful aa conditions mTORC1 is active and it in turn activates several different downstream targets leading to protein synthesis and cell growth. When amino acids are scarce, mTORC1 becomes inactive and this leads to a slow-down in protein synthesis and growth and an induction of autophagy, a pathway that generates nutrients from self-digestion of cellular material (*Gulati and Thomas, 2007*; *Kim et al., 2009*; *Chang et al., 2009*; *Wang and Proud, 2009*).

The mechanism by which amino acids are sensed by mTORC1 is beginning to be elucidated (reviewed in *Laplante and Sabatini, 2012*; *Jewell and Guan, 2013*; *Bar-Peled and Sabatini, 2014*). It appears that the active form of mTORC1 that responds positively to amino acid availability resides on late endosomal/lysosomal membranes, whereas absence of amino acids causes the translocation of mTORC1 from this compartment into the cytosol. Two protein complexes are responsible for the localization of mTORC1 to late endosomal/lysosomal membranes: a heterotetrameric complex of the RAG GTPases and a multimeric complex termed RAGULATOR, both of which are present on the late endosomal/lysosomal compartment constitutively (*Klm et al., 2008*; *Sancak et al., 2008*, *2010*). Activation state of the RAGs is partially determined by the RAGULATOR acting as a nucleotide exchange factor (*Bar-Peled et al., 2012*) and by an additional complex known as the GATOR acting as a GTPase activating protein (*Bar-Peled et al., 2013*) although it is also possible to activate mTORC1 downstream of amino acids in a way that is independent of the RAGs but still sensitive to the vacuolar ATPase (*Jewell et al, 2015*). In addition to the fundamental role of amino acids acting via the RAG/RAGULATOR axis, a small GTPase termed RHEB is also essential for mTORC1 activation

**eLife digest** Cells in all organisms must constantly measure the amount of nutrients available to them in order to be healthy and grow properly. For example, cells use a complex sensing system to measure how many amino acids – the building blocks of proteins – are available to them. One enzyme called mTOR alerts the cell to amino acid levels. When amino acids are available, mTOR springs into action and turns on the production of proteins in the cell. However, when amino acids are scarce, mTOR turns off, which slows down protein production and causes the cell to begin scavenging amino acids by digesting parts of itself.

Studies of mTOR have shown that the protein cannot turn on until it visits the surface of small sacks in the cell called lysosomes. These are the major sites within cell where proteins and other molecules are broken down. Scientists know how mTOR gets to the lysosomes, but not how quickly the process occurs.

Now, Manifava, Smith et al. have used microscopes to record live video of the mTOR enzyme as it interacts with amino acids revealing the whole process takes place in just a few minutes. In the experiments, a fluorescent tag was added to part of mTOR to make the protein visible under a microscope. The video showed that, in human cells supplied with amino acids, mTOR reaches the lysosomes within 2 minutes of the amino acids becoming available. Then, within 3-4 minutes the mTOR turns on and leaves the lysosome. Even though the mTOR has left the lysosome, it somehow remembers that amino acids are available and stays active.

The experiments show that mTOR's brief interaction with the lysosome switches it on and keeps it on even after mTOR leaves. Future studies will be needed to determine exactly how mTOR remembers its interaction with the lysosome and stays active afterwards.

(**Dibble and Manning, 2013**). This appears to be achieved via the amino acid-dependent translocation of the GTPase activating protein for RHEB termed TSC2 to the lysosomal surface (**Demetriades et al., 2014**).

A fundamental question of whether the primary amino acid sensor(s) are at the plasma membrane or inside the cell is not completely answered (**Dann and Thomas, 2006**; **Goberdham and Boyd, 2009**; **Hundal and Taylor, 2009**). The majority of recent work concludes that the sensors are internal, suggesting that amino acid transport to the cell interior is critical. A model proposed by Murphy and colleagues describes a coupled pathway whereby L-glutamine is first imported, then rapidly exported with the simultaneous import of L-leucine and other essential amino acids leading to mTORC1 activation (**Nicklin et al., 2009**). Other recent work suggests that cytosolic leucyl-tRNA synthetase is a leucine sensor and activates mTORC1 (**Han et al., 2012**; **Bonfils et al., 2012**), consistent with the known primacy of leucine among amino acids in activation of mTORC1. An "inside-out" sensing mechanism has also been proposed whereby amino acids in the lysosomal lumen are sensed by the vacuolar ATPase, and this signal is transmitted to the lysosomal exterior for activation of the RAGULATOR (**Zoncu et al., 2011**). All of these data support the idea that amino acids ultimately activate mTORC1 in the cell interior, and probably more than one type of sensing (and amino acid) is involved. For example, although leucine is sensed by the Sestrin1/2 proteins (**Wolfson et al., 2016**), recent work demonstrates that levels of arginine entering the lysosomes via transporter SLC38A9 are also important for mTORC1 activation (**Rebsamen et al., 2015**; **Wang et al., 2015**). At the same time, arginine is also sensed by the CASTOR1/2 proteins in the cytosol (**Chantranupong et al., 2016**) suggesting that even for the same amino acid multiple sensors in distinct cellular sites may be involved. On the other hand, it should also be mentioned that some older experiments showed that non permeable amino acid analogues can inhibit autophagy (**Miotto et al., 1992**, **1994**), suggesting that sensing may also involve events at the plasma membrane.

The exact spatial and temporal dynamics of the activation of mTORC1 upon amino acid stimulation are not understood owing to the difficulty of generating tagged versions of the kinase amenable to live imaging that still maintains physiological localization characteristics. In addition, making stable cell lines with other mTORC1 components such as RAPTOR also results in dynamics that differ

significantly from the endogenous complex (unpublished results). We have tried to overcome this problem in two different cell lines and using two different strategies. In HEK-293 cells, we have down-regulated endogenous RAPTOR while simultaneously expressing exogenous tagged constructs; in the near haploid HAP-1 cells we have tagged the endogenous RAPTOR gene with GFP. We describe here the dynamics of activation of the mTORC1 complex in these cells, with special emphasis in the HAP-1 cells. To examine whether mTORC1 responds directly to lysosomal amino acids, we have also generated a cell permeant fluorescent analogue of leucine in order to image RAPTOR translocation in response to this compound.

## Results

### Amino acid-dependent mTOR localization in fixed cells

We compared the timing of the translocation of the endogenous mTOR to a punctate compartment in response to amino acids with the phosphorylation of S6K, one of mTORC1's main substrates (*Figure 1A–C*). As previously reported (*Sancak et al., 2008*), in fed cells mTOR was partially in LAMP-1-positive punctate structures but became cytosolic during starvation from growth factors and amino acids (B). Upon re-stimulation with just amino acids, mTOR was found in a punctate distribution with a peak at 5 min (B, C). In a parallel experiment, mTOR phosphorylated S6K with a peak at 10 min of amino acid re-stimulation and phosphorylation continued to be strong even at 20 min (A). The discrepancy in the timing of mTOR translocation versus activity could be a natural delay until the complex becomes fully competent for phosphorylation. We noted however that in time points where the intensity of mTOR staining on lysosomes was back to low levels, phosphorylation of S6K was still strong (20 min), indicating that the pool of mTOR undertaking phosphorylation may not have been fully on the lysosomes. We also examined the localization of the direct mTORC1 target 4EBP1 as well as the S6K target the S6 protein under these conditions using antibodies that stain the endogenous phosphorylated proteins. The phosphorylated form of 4EBP1 did not localize on LAMP-1-positive lysosomes upon amino acid stimulation where activated mTOR would be expected to reside (*Figure 1*; *Figure 1—figure supplement 1A–C*). Similarly, the phosphorylated form of S6 was also not localised to LAMP-1-positive lysosomes at any time during recovery (*Figure 1*; *Figure 1—figure supplement 1D,E*).

An important protein in mTOR signalling is RHEB, a small GTPase absolutely required for mTOR activity (*Dibble and Manning, 2013*). Current models suggest that RHEB and the mTOR complex interact on the lysosomes for subsequent mTOR activation. In our hands, overexpressed RHEB capable of activating mTOR independently of the amino acid presence was primarily localised to the ER and Golgi and co-localized very weakly with endogenous mTOR (*Figure 1*; *Figure 1—figure supplement 2A,B*). In addition, endogenous RHEB co-localized very weakly with the lysosomal pool of mTOR under fed conditions or upon a round of starvation and re-stimulation where interaction with mTOR would be expected to be maximal (*Figure 1*; *Figure 1—figure supplement 2C*). In fact, a substantial amount of immunostained RHEB in these cells co-localized with the Golgi protein Giantin, and its localization was sensitive to Brefeldin A (*Figure 1*; *Figure 1—figure supplement 2D*), a drug that causes Golgi disappearance into the endoplasmic reticulum (*Lippincott-Schwartz et al., 1989*). An important regulator of RHEB activity is the TSC complex which inactivates RHEB by accelerating GTP hydrolysis, and it has been reported that the TSC2 protein cycles on and off the lysosomal surface in response to amino acid levels: when amino acids are withdrawn TSC2 translocates to the lysosomal surface where it inactivates RHEB causing the subsequent mTOR inactivation (*Demetriades et al, 2014*). Although we could verify this amino acid dependent translocation of TSC2 in HeLa cells, in our HEK-293 cells this appeared to be either less evident or even in the opposite way (more perinuclear in fed cells and less in starved, *Figure 1*; *Figure 1—figure supplement 2E*). Of note, mTORC1 activation dynamics in response to amino acids are very comparable in HEK-293 and HeLa cells (data not shown). These results suggested that the lysosome is a crucial locus for mTOR activation but the dynamics of this process and how mTORC1 activity is connected to lysosomal residence may be more complicated than current models.

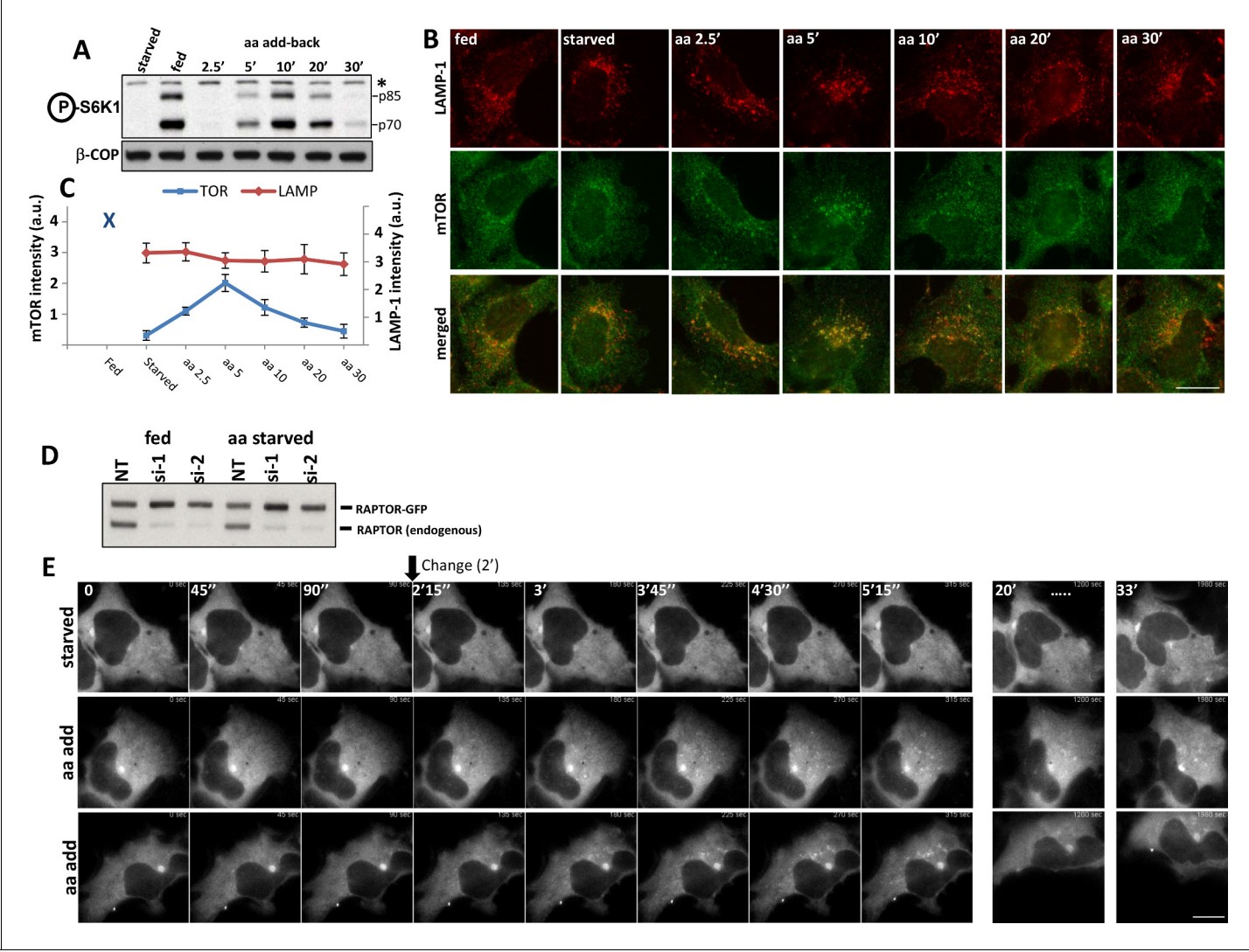

**Figure 1.** Activation of mTOR components in response to amino acids. HEK-293 cells were starved of amino acids and growth factors, and re-stimulated for the indicated times with MEM amino acids. (**A**) The phosphorylation of S6K at T389 was established as a surrogate for mTORC1 activity. Band marked with an asterisk serves as loading control. (**B**) Under the same conditions the localization of mTOR and the LAMP-1 lysosomal protein were examined by indirect immunofluorescence using antibodies that recognise the endogenous protein. (**C**) The total intensity of mTOR- or LAMP-1-positive puncta for the conditions shown in (**B**) were quantitated and plotted. We used 2 biological replicates with 10 technical repeats (10 regions of interest) and the error bars represent standard error of the mean. (**D**) HEK-293 cells were co-transfected with siRNAs targeting the 3' (si-1) and the 5' (si-2) untranslated region of RAPTOR together with plasmids expressing RAPTOR-GFP for 3 days. Levels of endogenous RAPTOR and the exogenous tagged protein were determined with RAPTOR antibodies. (**E**) Live cell imaging of samples as in **D**. Cells were starved of amino acids and growth factors and re-stimulated either with starvation medium (starved) or with amino acids (aa add) for the indicated times. Representative frames are shown. The change to either starvation or amino acid medium was done after 90'' of incubation. The bar in all panels represents 10 µm.

The following figure supplements are available for figure 1:

**Figure supplement 1.** Localization of mTOR-phosphorylated 4EBP1 and S6K-phosphorylated S6 in response to amino acid stimulation.

**Figure supplement 2.** Localization of overexpressed and endogenous RHEB in relation to mTOR.

**Figure supplement 3.** Localization and activity of RHEB upon Golgi disruption.

## Dynamics of RAPTOR-GFP in HEK-293 cells

To examine the dynamics of mTOR in response to amino acids we tried to express subunits of mTORC1 tagged with GFP in order to follow localization in live cells. Tagging either mTOR or RAPTOR with GFP at either N' or C' terminus and generating either transient or stable cell lines of varying expression levels did not provide us with model systems that recapitulated the dynamics of endogenous mTOR (data not shown). One approach that was partially successful was to transfect cells with siRNAs targeting the 3' and 5' untranslated regions of endogenous RAPTOR while simultaneously expressing transiently in the same cells RAPTOR-GFP (*Figure 1D*). In these cells, endogenous RAPTOR expression was greatly reduced and exogenous expression (resistant to the siRNAs) could be titrated to be similar to the endogenous. In live imaging experiments using these cells after a round of starvation (removing amino acids and growth factors) and re-stimulation (adding back amino acids) we observed a rapid translocation of RAPTOR-GFP to small and fine punctate structures within 1 to 2 min of amino acid addition (*Figure 1E*, panels labelled "aa add"). In contrast, re-stimulation of these cells with starvation medium alone did not reveal a translocation of RAPTOR-GFP (*Figure 1E*, panel labelled "starved"). Although this was a reproducible effect, it was evident in a minority of the cells in the population, primarily in those expressing very low levels of exogenous RAPTOR-GFP. This cell variability made it very difficult to prove by biochemical methods that mTOR was being activated and we did not pursue this line of experimentation further.

## Generation of HAP-1 cells expressing RAPTOR-GFP from the endogenous locus

An alternative approach was to introduce to the endogenous RAPTOR a GFP tag. For this we used HAP-1 cells which are derived from the near haploid cell line KBM-7 via an unsuccessful attempt at reprogramming (*Carette et al., 2011*; *Burckstummer et al., 2013*). Using a CRISPR/Cas9 approach we derived a clone that expressed RAPTOR-GFP from the endogenous locus and at equal levels to the parental cells (*Figure 2A*). The fluorescence signal in these cells was sufficiently bright to be discerned by wide field microscopy (*Figure 2B*) although not strong enough for long illumination in confocal microscopy (data not shown). The levels of endogenous mTOR in these cells were equal to the wild type cells (*Figure 2A,C*) and the RAPTOR-mTOR complex could be immunoprecipitated with antibodies to endogenous mTOR (*Figure 2C*) using the lysis conditions in 0.3% CHAPS previously described (*Kim et al., 2002*). In order to ensure that these cells could be used for live imaging in response to amino acid stimulation we characterised their mTORC1 signalling properties using as surrogates for mTORC1 activity the phosphorylation of two direct downstream targets: S6K and ULK1. In fed conditions the RAPTOR-GFP cells exhibited a slightly lower phosphorylation of S6K than the HAP-1 parental cells (*Figure 2D*). Both cells showed an inhibition of S6K and ULK1 phosphorylation upon starvation (removing amino acid and growth factors) and a comparable re-stimulation with their full growth medium (Iscove's modified Dulbecco's medium-IMDM *Figure 2D*, imdm10' and imdm20'). Interestingly, upon re-stimulation with MEM amino acids the RAPTOR-GFP cells were significantly less responsive than the parental ones [*Figure 2D*, aa(MEM)10' and aa(MEM) 20']. This was further explored by combining MEM and non-essential (NE) amino acids during re-stimulation. Whereas this made only a small difference to the parental cells (*Figure 2E*, HAP-1 lanes) it did significantly increase the response in the RAPTOR-GFP cells for both S6K and ULK1 phosphorylation (*Figure 2E*, lanes labelled "HAP-1 RAPTOR-GFP"). Of note, NE amino acids on their own did not activate mTOR in either cell line. From this point on, an "aa" designation for the HAP-1 RAPTOR-GFP cells will indicate a mixture of MEM and NE amino acids. Although a combination of amino acids activated mTOR to reasonable levels in the RAPTOR-GFP cells, this was not as high as in parental cells, or in RAPTOR-GFP cells stimulated with full growth medium. To overcome this, we added during re-stimulation two growth promoting factors, Insulin and EGF, and examined mTOR activation. These two growth factors, and especially Insulin, had a further enhancing effect although on their own they did not activate mTOR (*Figure 2F*, blot labelled 1% dial serum). All of these starvation and re-stimulation experiments were done in medium containing salts and 1% dialysed serum. When we repeated them in medium containing 1% BSA instead of dialysed serum we saw that in the parental cells re-stimulation was still largely dependent on amino acids and weakly on added growth factors whereas re-stimulation in the RAPTOR-GFP cells followed the requirements as above (weak with amino acids, further enhanced with growth factors) but it was to much lower final levels

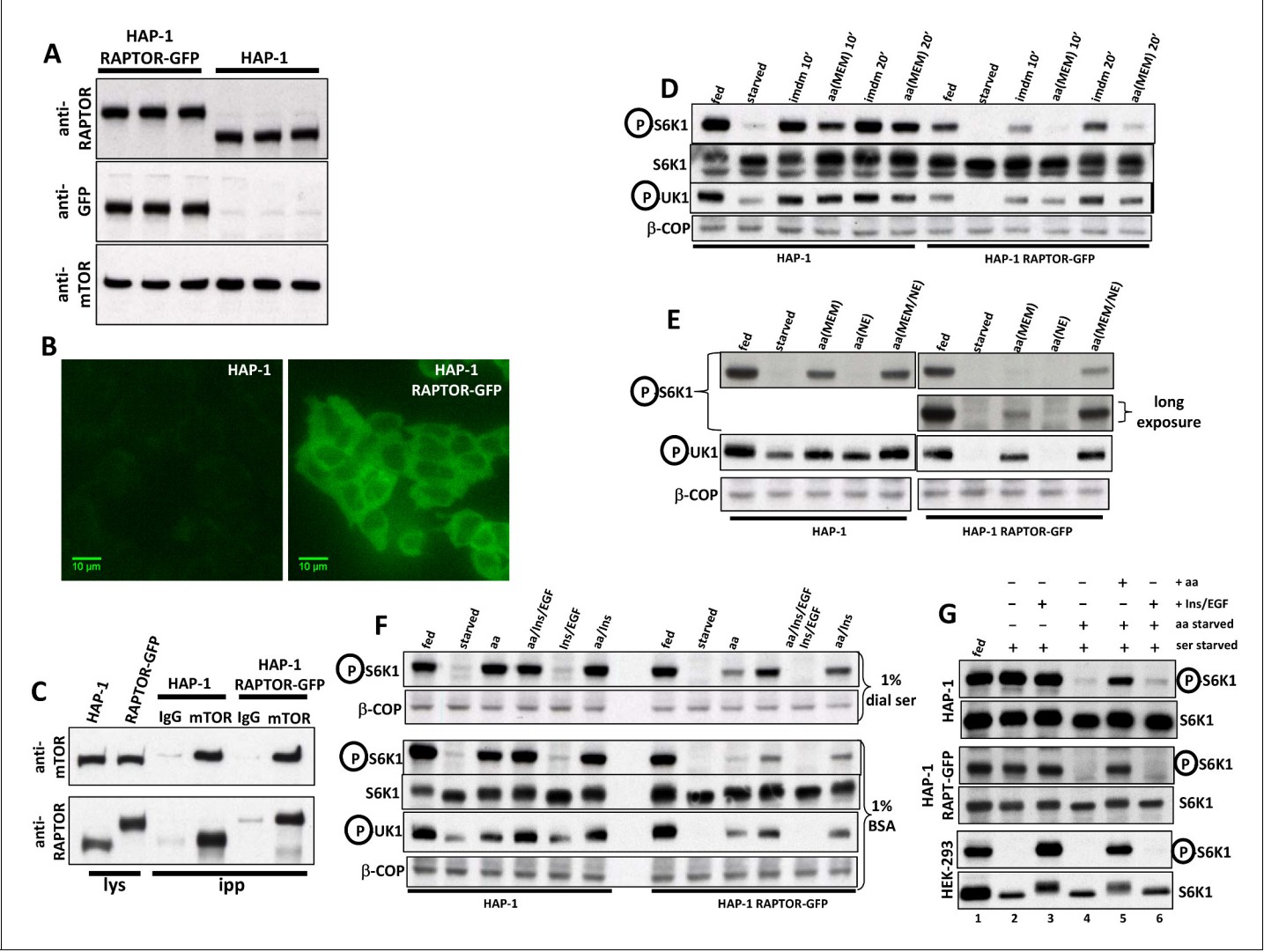

**Figure 2.** Characterization of HAP-1 cells expressing RAPTOR-GFP in place of endogenous RAPTOR. (**A**) Immunoblots of parental HAP-1 cells or HAP-1 cells expressing RAPTOR-GFP, in triplicate. (**B**) Fluorescence microscopy of cells as in A; both photos were taken at the same setting of brightness and similarly edited in Photoshop. The bar represents 10 μm. (**C**) Cells as in A were lysed in 0.3% CHAPS lysis buffer and lysates subjected to immunoprecipitation using either mTOR antibodies or antibodies to PAK1 used as an IgG control. After immunoprecipitation and electrophoresis the samples were blotted with antibodies to RAPTOR or to mTOR. (**D**) Cells as in A were kept in normal medium (Iscove's modified Dulbecco's medium - IMDM plus 10% FBS) or starved of amino acids and growth factors for 60 min in medium containing salts and 1% dialysed FBS. The cells were then re-stimulated for 10' or 20' either with normal growth medium (imdm) or with a mixture of MEM amino acids added to the starvation medium (aa). After lysis and electrophoresis the samples were immunoblotted for the indicated proteins. (**E**) Cells kept fed or starved as above were re-stimulated for 20 min with starvation medium containing a mixture of MEM amino acids, a mixture of non-essential (NE) amino acids or a combination of both. Detection was for S6K phospho T389 or ULK1 phospho S757. (**F**) Cells as in A were starved of amino acids and growth factors in medium containing salts and either 1% dialysed FBS (top panel) or 1% BSA (bottom panel). Re-stimulation for 20' was in medium containing either dialysed FBS or BSA matching the starvation condition and also containing amino acids (mixture of MEM and NE), amino acids plus Insulin and EGF, Insulin and EGF only or amino acids plus Insulin. Detection was for the indicated proteins. (**G**) Parental HAP-1 cells, HAP-1 cells expressing RAPTOR-GFP or HEK-293 cells as indicated were starved of serum overnight. Next day, some samples were also starved of amino acids for 60' as indicated and then re-stimulated for 20' in medium containing growth factors (Insulin and EGF) and/or amino acids. Detection was for S6K or S6K phospho T389.

compared to the fed condition (*Figure 2F*, blot labelled 1% BSA). We also examined the response of the two HAP-1 cells and the HEK-293 cells to serum starvation on its own or in combination with a subsequent amino acid starvation. The activity of mTORC1 in both HAP-1 cells was remarkably resistant to overnight serum withdrawal whereas in HEK-293 cells this treatment completely inhibited

mTORC1-directed phosphorylation of S6K (*Figure 2G*, first two lanes in each cell line). Consequently, stimulation of the serum starved cells with growth factors only stimulated HEK-293 cells and not the two HAP-1 cells (*Figure 2G*, third lane). Importantly, a subsequent amino acid starvation regime in all three cell lines followed by amino acid re-addition showed the expected inhibition and re-stimulation of mTORC1 (*Figure 2G* lanes 4 and 5), whereas growth factors on their own under these conditions did not re-activate mTORC1 (*Figure 2G*, lane 6). We concluded from all of this characterization that the RAPTOR-GFP cells can be used for analysing the dynamics of amino acid stimulation of mTORC1, but that the presence of RAPTOR-GFP in place of the endogenous RAPTOR makes the cells overall more dependent on additional factors (other than amino acids and present in serum) in order to obtain a maximal response. However, it should emphasised that in the RAPTOR-GFP cells it is the MEM amino acids that are driving mTORC1 activation even in the presence of the additional factors whereas serum withdrawal on its own has a minimal effect on mTORC1.

## Dynamics of translocation in response to amino acids

The HAP-1 cells are not ideal for immunofluorescence analysis because they have a large nucleus and a small cytosol. Nevertheless when we examined the RAPTOR dynamics in these cells in response to amino acid starvation and re-stimulation we saw that, whereas the protein was primarily cytosolic in starved cells, it translocated to punctate structures upon amino acid stimulation (*Figure 3*). Importantly, these RAPTOR-GFP puncta co-localised with endogenous mTOR (*Figure 3A,C*) or RAGC (*Figure 3B,D*), indicating that the RAPTOR-GFP protein was localizing with its endogenous interacting partners. As expected, whereas mTOR was cytosolic during starvation and became punctate upon stimulation, RAGC was continuously present on punctate structures as reported before for other cell types. These punctate structures to which RAPTOR-GFP and mTOR translocated were lysosomes as revealed by co-localization with the lysosomal protein LAMP-1 (*Figure 3C,D* and see below).

We then set out to examine the RAPTOR-GFP dynamics in these cells by live imaging. After a round of amino acid starvation and re-stimulation, the protein very rapidly translocated to puncta that were also positive for the co-expressed mRFP-LAMP-2 lysosomal protein (*Figure 4A* and *Video 1*). This translocation of RAPTOR-GFP to the lysosomal compartment was evident within 2 min of amino acid addition, very reminiscent of the dynamics that we saw in HEK-293 cells (*Figure 1D*), and it was substantially reduced by about 10 min of stimulation. Throughout the peak of translocation, RAPTOR-GFP was significantly co-localized with overexpressed mRFP-LAMP2 (*Figure 4A*, plot) or with endogenous LAMP-1 (*Figure 4B*). The dynamics of co-localization as measured by estimating a Pearson's coefficient suggest that soon after aa stimulation co-localization is 3-fold higher than basal levels, which returns close to baseline at later times (*Figure 4A*, plot). Of note, the localization of mRFP-LAMP-2 or of LAMP-1 was unaffected throughout the stimulation, indicating that what we observed was translocation of RAPTOR to pre-existing lysosomes.

An important recent observation was that amino acid-dependent mTOR stimulation is sensitive to inhibitors to the vacuolar ATPase (*Zoncu et al., 2011*). Indeed, when cells were starved and re-stimulated with amino acids in the presence of concanamycin A, a potent vacuolar ATPase inhibitor, the translocation of RAPTOR-GFP to puncta was largely inhibited (*Figure 4C*, *Video 2* and *Figure 5C, D*).

Translocation of RAPTOR-GFP to punctate structures such as shown in *Figure 4C* and *Video 2* was robust and reproducible enough to allow us to plot translocation as a function of time after amino acid addition (*Figure 5B,D*). It is apparent that the response is very fast (evident within 2 min of amino acid addition) and returns to basal levels also quickly. In parallel experiments and using exactly the same media and cells on the same day we also determined in triplicate the activation of mTORC1 by biochemical methods (*Figure 5A*). Here we saw that phosphorylation of S6K was barely evident after 10 min of amino acid stimulation and continued to increase even at 30 min post stimulation. We will address this difference in the dynamics of translocation and activation at a later section. As discussed above, we also determined that the dynamics of RAPTOR-GFP translocation were sensitive to inhibition of the vacuolar ATPase using concanamycin A, and it agreed with the effects of this compound on mTOR activity (*Figure 5C,D*). In addition to its strong effects on the dynamics of translocation, concanamycin A also reduced by approximately 40% the number of cells that showed any response at all (*Figure 5E,F*). Finally, we compared directly the dynamics of translocation of RAPTOR-GFP in the presence of amino acids alone or amino acids plus EGF and insulin,

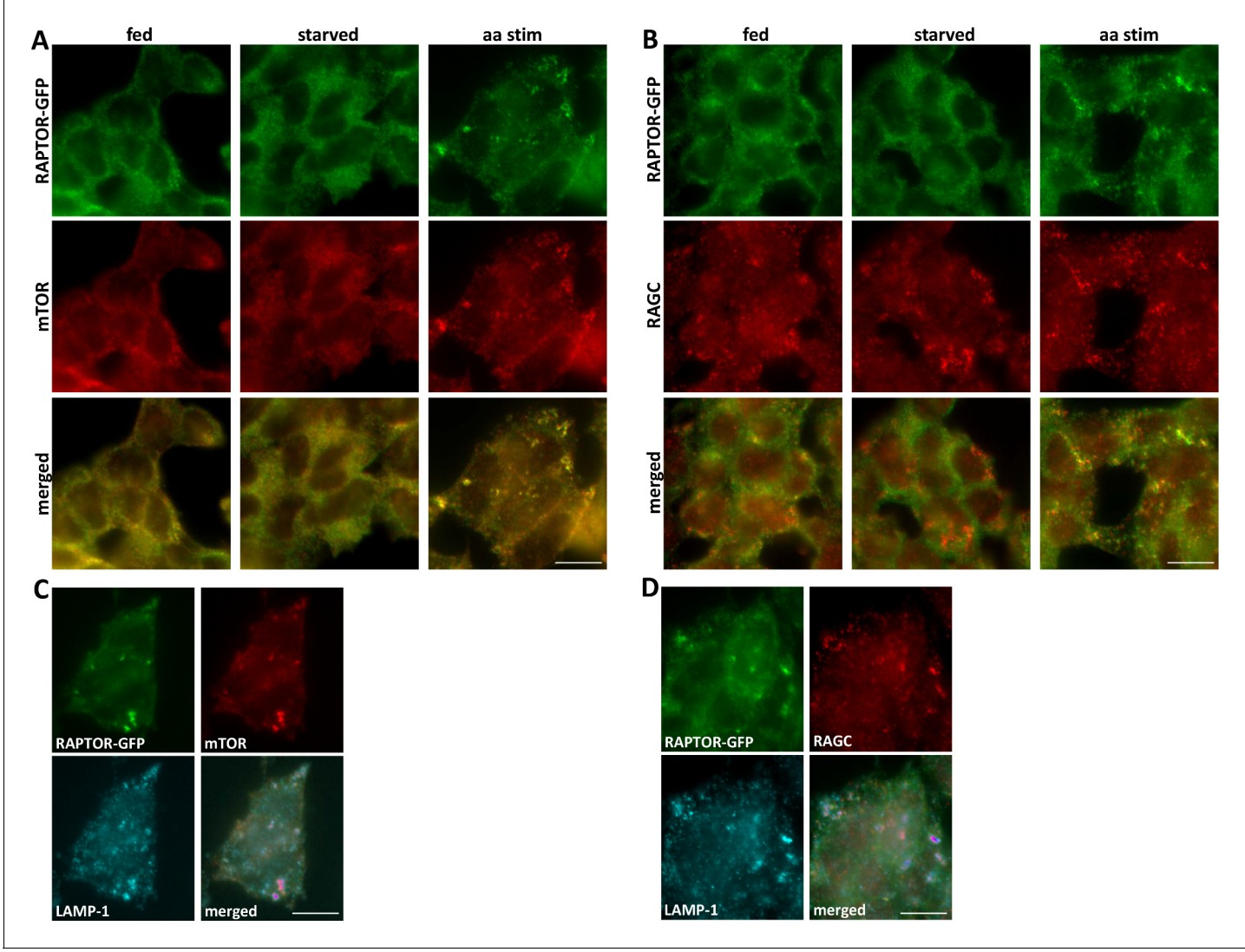

**Figure 3.** Translocation of RAPTOR-GFP to mTOR- and RAGC-positive compartments in response to amino acid stimulation. (A, B). HAP-1 cells expressing RAPTOR-GFP were kept under fed conditions or were starved of amino acids and growth factors and re-stimulated with a mixture of MEM and NE amino acids for 15'. The cells were fixed and stained with antibodies to endogenous mTOR (A) or RAGC (B). Detection was for GFP (RAPTOR) or using red secondary antibodies for mTOR and RAGC. (C, D). Cells starved and re-stimulated as above were fixed and simultaneously stained for mTOR and LAMP-1 (C) or RAGC and LAMP-1 (D). The bar in all panels represents 10 µm.

conditions that enhance the phosphorylation response as shown in *Figure 2F*. In these experiments we did not see a difference in the dynamics of the response (*Figure 5D* and *Video 3*) but we did observe that a higher proportion of the cells in our fields of view showed translocation when growth factors were added (*Figure 5F*). We concluded from this that growth factors do not change the dynamics of the response but rather expand the number of cells capable of showing a response. RHEB is the protein that appears to integrate growth factor signalling to mTOR (*Dibble and Manning, 2013*); we hypothesise that the presence of GFP at the C terminus of RAPTOR may make the complex more dependent on growth factor inputs during amino acid stimulation. However, we emphasise that mTORC1 activation in these cells still depends on amino acids.

## Sustained mTORC1 activity in comparison to its translocation dynamics

The observed kinetic difference between RAPTOR-GFP translocation and phosphorylation of its substrates suggests either that mTOR activity persists long after the complex has come off the

lysosomal surface or that de-phosphorylation of S6K continuously antagonises the phosphorylation step in a way that does not allow the phosphorylated protein to accumulate at the early time points. To address the second possibility directly we examined the rate of dephosphorylation of S6K after a round of stimulation for different time points. First cells were starved and then re-stimulated with amino acids and growth factors for either 20 min or 50 min, at which point a mTOR inhibitor was added to block any further phosphorylation. The cells were then allowed to incubate for different times before phosphorylation of S6K was measured by immunoblots. In those experiments we saw that phosphorylation of S6K was rapidly lost within 5 min of incubation with the mTOR inhibitor and the dynamics of this were identical for the early and late points after stimulation (*Figure 5*; *Figure 5—figure supplement 1A,B*). To investigate further the state of mTORC1 interactors during a round of starvation and re-stimulation we used immunoprecipitation of RAPTOR from these cells using the GFP tag (*Figure 5G*). In the 0.3% CHAPS lysis buffer, a substantial amount of the mTOR/ RAPTOR complex can be isolated from fed cells and the stoichiometry of this complex is not altered during starvation and recovery (*Figure 5G*). We did not observe binding under our conditions of RHEB, ULK1 and ATG13, the latter two being autophagy proteins downstream of mTORC1 signalling. In contrast, we saw differential binding of RAGA to the complex: the protein was present in fed conditions, greatly reduced in starved cells and then binding was again evident until 20 of re-stimulation only to drop again afterwards. This was seen very reproducibly and with two independent conditions of lysis (*Figure 5G*, panels labelled RAGA and RAGA*) and it provides further evidence that when mTORC1 activity is still very strong (for example at 30 min of re-stimulation, see *Figure 5A,C*) the interaction of mTOR with its known regulators is diminished.

## Translocation in response to a fluorescent amino acid analogue

We also addressed directly the temporal aspect of RAPTOR-GFP translocation in response to amino acid stimulation. Among amino acids, leucine has a prominent role in mTORC1 activation, and it has also been reported that a leucine methyl ester that enters cells and accumulates in lysosomes can activate mTORC1 on its own (*Zoncu et al., 2011*). Prompted by this, we tested a series of leucine analogues for mTORC1 activation (*Figure 6*; *Figure 6—figure supplement 1A,B*). In our hands, a di-leucine methyl ester was a potent activator of mTORC1, much stronger that the single leucine methyl ester which was in turn much stronger than leucine itself (*Figure 6*; *Figure 6—figure supplement 1C*). Interestingly, a methyl ester of L-L-G was also a very strong activator of mTORC1 (*Figure 6*; *Figure 6—figure supplement 1C*). On this basis we synthesised a compound consisting of a double leucine followed by the Mant fluorophore, one of the smallest fluorescent molecules (*Figure 1*; *Figure 1—figure supplement 1A*, fluorescent analogue). On its own, this analogue could activate mTORC1 after a round of starvation and re-stimulation at levels comparable to a mixture of NEM amino acids, albeit slightly slower but with the same sensitivity to the vacuolar ATPase inhibitor concanamycin A (*Figure 6*; *Figure 6—figure supplement 1D*). In addition, this analogue entered cells instantaneously upon addition, and targeted primarily the lysosomes in a way not inhibited by concanamycin A (*Figure 6*; *Figure 6—figure supplement 1E*). The fluorescent analogue was then used in the HAP-1 cells expressing RAPTOR-GFP. Here we found that whereas in the parental cells both the di-leucine methyl ester as well as the fluorescent analogue were capable of activating mTORC1 in medium containing growth factors and NE amino acids (*Figure 6A*, last two lanes of HAP-1 cells), this was much reduced in the cells expressing RAPTOR-GFP, once again suggesting that the presence of the transgene makes the cells dependent on a more complex activation mixture. Based on recent observations that arginine may be an equally important amino acid to leucine for mTORC1 activation (*Rebsamen et al., 2015*; *Wang et al., 2015*), we then stimulated the RAPTOR-GFP cells with a mixture of arginine and the fluorescent analogue (*Figure 6B*). This condition proved to be favourable for activating mTORC1 [lanes labelled aa(NE)/GF/FA/R in *Figure 6B*]. Of note this activation, although it required all of the other components (growth factors, NE amino acids and arginine), it was absolutely dependent on the presence of the fluorescent analogue. We then used live imaging to examine the dynamics of RAPTOR-GFP translocation in response to the fluorescent analogue. We documented a very reproducible sequence of events whereby the analogue first entered lysosomes and, very shortly, RAPTOR-GFP started to translocate there (*Figure 6C*, and *Video 4*). Interestingly, although we never saw RAPTOR translocation to lysosomes devoid of the analogue, not all labelled lysosomes became RAPTOR positive. In the majority of cases we saw that the strongest-staining organelle became RAPTOR-positive first (example in *Figure 6D*),

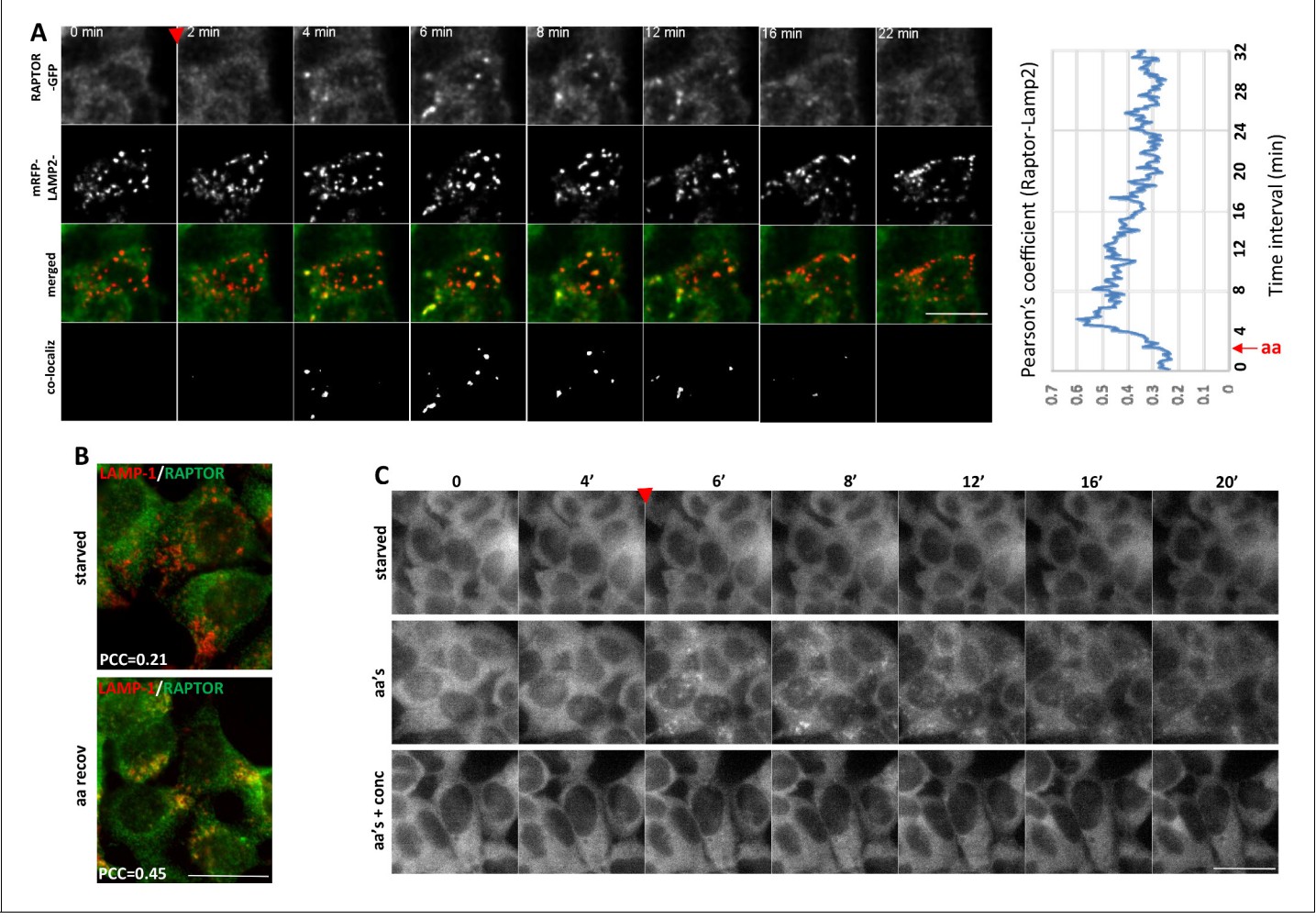

**Figure 4.** Translocation of RAPTOR-GFP to lysosomes and its dynamics in response to amino acid stimulation. (**A**) HAP-1 cells expressing RAPTOR-GFP were co-transfected with plasmids expressing mRFP-LAMP2 for 24 hr. The cells were then starved of amino acids and growth factors for 60' and assembled on an imaging chamber still in starvation medium. Live imaging was started in a starvation medium for 2' followed by replacement with medium containing a mixture of MEM and NE amino acids for an additional 40'. Images were captured every 10''. Shown here are frames from such a movie in the two channels (GFP and red) the merged images and the co-localizing areas for the indicated time points. The extent of co-localization between RAPTOR-GFP and mRFP-LAMP2 as determined by the Pearson's coefficient is plotted on the graph. Images were background-subtracted using the rolling ball method in FIJI (*Schindelin et al., 2012*) with a diameter of 10 pixels. Co-localization analysis was performed using Imaris software (Bitplane/Oxford Instruments) with thresholds for co-localization set using the auto function to avoid user bias. See also *Video 1*. (**B**) HAP-1 cells expressing RAPTOR-GFP were starved of amino acids and growth factors, and re-stimulated for 20'. The cells were stained for GFP (RAPTOR) and endogenous LAMP-1, and the extent of co-localization was determined by the Pearson's co-efficient (PCC) shown. (**C**) HAP-1 cells expressing RAPTOR-GFP were starved of amino acids and growth factors for 60' and set up for live imaging as above. Re-stimulation after 5' of imaging was in the starvation medium (starved), in the starvation medium containing a mixture of MEM and NE amino acids (aa's), or in medium containing amino acids and 2 µM concanamycin A (aa's + conc) for an additional 40'. Images were captured every 10''. Selected frames are shown. Note that in the concanamycin A-treated samples we found it necessary to pre-treat the cells for the last 10' of starvation with the compound before adding it again during re-stimulation. The bar in all panels represents 10 µm.

but we also saw examples where several lysosomes containing analogue became RAPTOR-positive simultaneously. In addition, we frequently saw that RAPTOR translocated to large vacuolar-like organelles containing the fluorescent analogue (example in *Figure 6E*). Amino acid methyl esters are known to continuously accumulate within lysosomes wherein the methyl ester moiety is hydro-lysed and amino acids are trapped (*Reeves 1979*); this provides a condition of continuous amino acid supply to provide sustained mTOR activation. To examine if this would also affect translocation, we quantitated translocation dynamics of RAPTOR-GFP in response to the fluorescent analogue

(*Figure 6F*). It was clear that translocation under these conditions was sustained (albeit starting more slowly than the response with amino acids) and the RAPTOR-GFP protein did not come off the lysosomes throughout our live imaging time interval (30 min). This result provides additional evidence that translocation of the complex to the lysosomal surface is in direct response to amino acid levels in the lysosomal lumen.

## Activation of mTORC1 by the amino acid analogue depends on Sestrin2 and on intact lysosomes

To our knowledge, the leucine analogue is the first fluorescent reagent capable of activating mTORC1 and at the same time visible by microscopy. In order to understand its mechanism of action further we set up parallel experiments in HEK-293 cells (where the analogue can activate mTORC1 completely on its own) and in the edited HAP-1 cells expressing RAPTOR-GFP (where activation is uniquely dependent on the analogue but also requires additional components). Given the recent work that Sestrin2 is a cytosolic sensor of leucine for mTORC1 activation (*Wolfson et al., 2016*) we determined if this protein is also involved in sensing the analogue. In both cell types, down-regulation of Sestrin2 increased mTORC1 activation by regular amino acids and by the fluorescent analogue (*Figure 7A and B*). In HAP-1 cells, reduction in Sestrin2 levels caused a 3- to 4-fold increase in mTORC1 activation at 10' and 20' of amino acid addition, with the process of activation reaching a plateau by 30 min (*Figure 7A*). For fluorescent analogue addition, the enhancement of mTORC1 activation was maintained at the longer time point (30') as well (*Figure 7B*), consistent with the idea that lysosomal translocation of RAPTOR in response to the analogue is more sustained than to amino acids (see *Figure 6F*). In HEK-293 cells, in addition to the significant enhancement of mTORC1 re-activation for the Sestrin2 siRNA samples, we also consistently saw that the starting levels of mTOC1 activity were significantly higher (*Figure 7B*, compare fed samples). In addition, HEK-293 cells down-regulated for Sestrin2 showed a significantly more enhanced response to the fluorescent analogue in comparison to the effects of regular amino acids (*Figure 7B*). All of these data are consistent with the idea that Sestrin2 is a negative regulator of GATOR2, which is itself a key negative regulator of GATOR1, and therefore reduction is Sestrin2 de-represses that signalling cascade resulting in enhanced mTORC1 activation (*Wolfson et al., 2016*). The finding that Sestrin2 reduction affects not only the starting levels of mTORC1 activity but the levels of its amino-acid induced reactivation is consistent with recent reports (*Peng et al, 2014*; *Ro et al, 2016*) and may indicate a dynamic equilibrium between Sestrin2 and GATOR2. Why does a lysosomally-targeted amino acid analogue still maintain dependence on Sestrin2 levels, a cytosolic sensor? One possibility is that a pool of analogue remains in the cytosol and never enters lysosomes. To address this possibility we asked whether intact lysosomes were important for the fluorescent analogue action by using GPN (glycyl-L-phenylalanine 2-naphththylamide) during recovery. GPN enters and accumulates in lysosomes leading to their rupture and has been historically used to implicate integrity of the organelles in several signalling pathways (*Berg et al., 1994*). In both HAP-1 RAPTOR-GFP and in HEK-293 cells, GPN significantly inhibited the ability of the fluorescent analogue to activate mTORC1 (*Figure 7C and E*) whereas its effects when amino acids were added were also evident (but less strong in the case of HAP-1 cells). Interestingly, whereas lysotracker staining of lysosomes was completely eliminated by GPN in both cell types (*Figure 7D and F*, top panels) the fluorescent analogue still accumulated (*Figure 7D and F*, bottom panels). Of note, activation of mTORC1 in response to the fluorescent analogue was also sensitive to vacuolar ATPase inhibitors for HEK-293 cells (*Figure 6*; *Figure 6—figure supplement 1D*) and for HAP-1 RAPTOR-GFP cells (not shown).

We also examined whether activation of mTORC1 by amino acids versus the fluorescent analogue could be mechanistically distinguished on the basis of sensitivity to macropinocytosis inhibitors based on recent observations that macropinocytosis of extracellular amino acids regulates mTORC1 activation (*Palm et al., 2015*; *Yoshida et al., 2015*). First, we showed that activation of mTORC1 via amino acids was sensitive to the macropinocytosis inhibitor 5-(*N*-ethyl-*N*-isopropyl) amiloride (EIPA), especially at later times after stimulation (*Figure 7*; *Figure 7—figure supplement 1A*) whereas the di-leucine methyl ester (LLOMe), which served as the basis for synthesising the amino acid analogue (see *Figure 6*, *Figure 1*), was insensitive to inhibition by EIPA (*Figure 7*, *Figure 7—figure supplement 1B*). Under the same conditions, stimulation by amino acids or LLOMe was sensitive to the vacuolar ATPase inhibitor concanamycin A (*Figure 7*, *Figure 7—figure supplement 1A and B*). The same differential sensitivity to EIPA was also seen for the fluorescent analogue itself such that

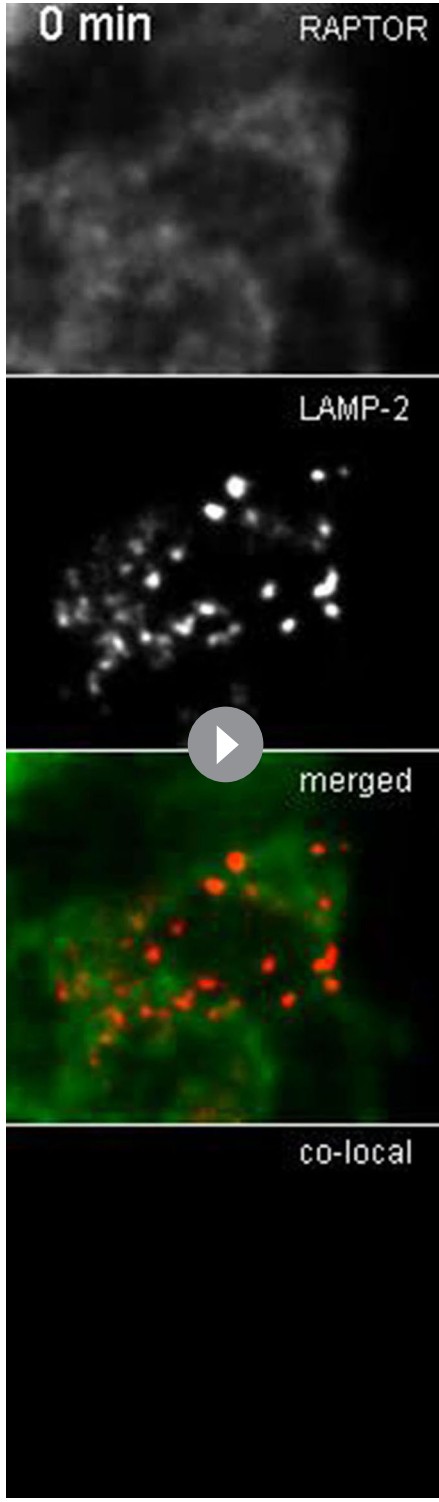

**Video 1.** Double imaging of RAPTOR-GFP and mRFP-LAMP2 in HAP-1 cells. After 55' of starvation in the incubator, cells were imaged for additional 5' of starvation on stage followed by on-stage addition of amino acid containing starvation medium. Display is at 15 frames per second. The last panel shows the co-localizing fraction which was used to derive a Pearson's coefficient plot. See also *Figure 4A*.

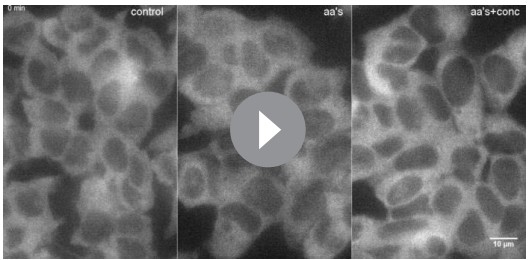

**Video 2.** Single imaging of RAPTOR-GFP HAP-1 cells. After 55 min of starvation in the incubator, cells were imaged for additional 5' of starvation on stage followed by on-stage addition of (1) starvation medium (2) amino acid containing starvation medium or (3) amino acid containing starvation medium plus 2 μM concanamycin A as indicated. Of note concanamycin A was also added for the last 10' of starvation. See also *Figures 4* and *5*. Display is at 15 frames per second.

stimulation of mTORC1 via the fluorescent analogue was insensitive to EIPA (*Figure 7*, *Figure 7—figure supplement 1C*).

These data suggest that the fluorescent analogue requires intact and functional lysosomes to signal to mTORC1 and that its mechanism of being sensed depends on Sestrin2 although it does not require delivery via macropinocytosis.

## Discussion

We examined the spatiotemporal dynamics of mTOR activation and its kinetic dependence on amino acid presence. Pioneering work from several laboratories had already provided strong evidence that mTOR itself translocates to lysosomes during amino acid re-stimulation (*Sancak et al., 2008*, *2010*; *Zoncu et al., 2011*), but this work had relied exclusively on a single antibody that can stain the endogenous protein and it did not provide any information on the dynamics of the process. The RAPTOR-GFP edited cells that we used here provide a unique tool to address this question in a near-physiological setting. Although they are slightly less responsive to stimuli than the parental cells and require a more carefully assembled mixture of activators, they nevertheless show absolute reliance on amino acids for mTORC1 activation and RAPTOR translocation to lysosomes. Our work shows that translocation is a very fast and relatively transient event that peaks early and then returns to base line, whereas phosphorylation of substrates under identical conditions and measured in parallel continues for some time

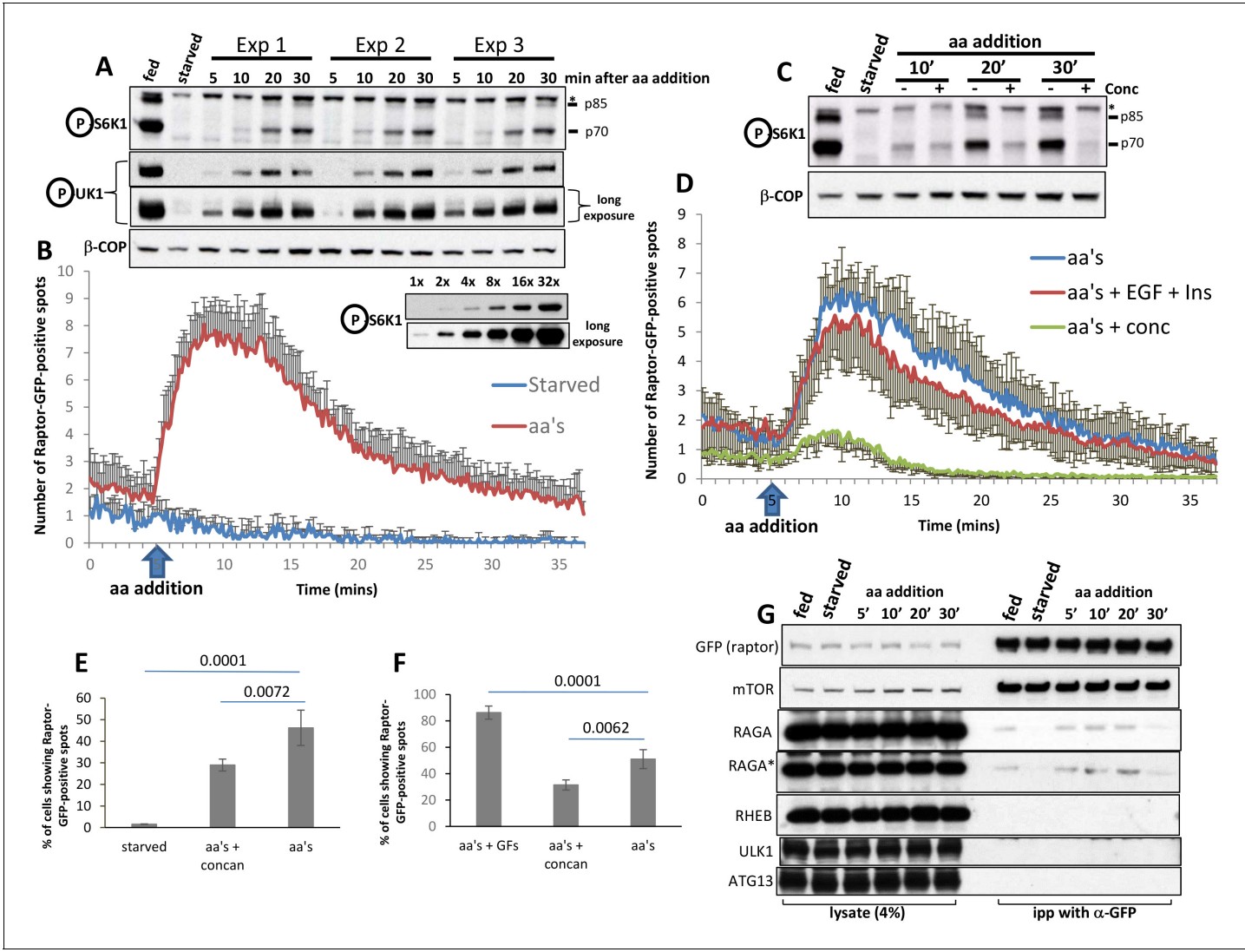

**Figure 5.** Dynamics of RAPTOR-GFP translocation to lysosomes in comparison to mTOR activity. HAP-1 cells expressing RAPTOR-GFP were set up for parallel experiments: live imaging to reveal RAPTOR dynamics (**B, D, E, F**) and immunoblotting to reveal S6K phospho T389 (**A, C**) and ULK1 phospho S757 (**A**). Immunoblotting was for the indicated times after re-stimulation with a mixture of MEM and NE amino acids. The linearity of the detection for S6K phospho T389 is shown for ascending amounts of lysate (last two blots in **A**). In the experiment shown in **A**, re-stimulation was in triplicate. In the blot shown in C, concanamycin A was added to a final concentration of 2 μM for the indicated samples. In order to obtain graphs of RAPTOR translocation to punctate structures (**B** and **D**) we examined many cells for each condition coming from 3 to 4 independent imaging experiments (13 cells for control and 137 cells for aa's in panel b; 59 cells for aa's, 77 cells for aa's + EGF + Ins and 60 cells for aa's + conc in panel d). The number of puncta upon stimulation was counted in each cell using the Spots Detection function of Imaris software (Bitplane/Andor), and the average of all cells in each condition together with the standard deviation is plotted here. The number of cells responding to the various treatments and expressed as a percentage is shown in panels **E** and **F**, together with the statistical significance of the differences as indicated. Significance was evaluated using one-way ANOVA with a Bonferoni post-hoc test. Error bars represent standard error of the mean. (**G**) HAP-1 cells expressing RAPTOR-GFP were kept in fed conditions, starved for 60' or starved for 60' and re-stimulated with amino acids and growth factors for 5', 10', 20' and 30'. The RAPTOR-GFP protein was immunoprecipitated from these cells and the immunoprecipitates were analysed for a variety of potential interacting proteins as shown. All experiments were conducted in 0.3% CHAPS lysis buffer with the exception of the one labelled RAGA* which was conducted in 0.3% deoxy big CHAP.

The following source data and figure supplement are available for figure 5:

**Source data 1.** Raw data file (excel) for *Figure 5* plots B and D.

**Figure supplement 1.** Dynamics of de-phosphorylation and a simple model.

thereafter. We have also demonstrated for the first time that the presence of an amino acid analogue in the lysosomal lumen is sufficient to induce the translocation of RAPTOR-GFP there. This analogue, a methyl ester of di-leucine with a fluorescent moiety attached, produces a sustained translocation of the complex presumably owing to its property to continuously accumulate within the lysosomal lumen. All of these data provide for the first time a dynamic view of mTORC1 activation in response to amino acids.

Despite tremendous progress, the exact mechanism by which mTORC1 senses amino acids appears very complex and incomplete at present (reviewed and commented on in: *Goberdham et al, 2016*; *Lee et al., 2016*; *Shimobayashi and Hall, 2016*). Clearly, several different sensors are involved (sometimes for the same amino acid, e.g. for arginine), and the topology of the phenomenon involves the cytosol as well as the lysosomal membrane and the lysosomal lumen. Our own data in two cell types argue that, in addition to intact lysosomes and the lysosomal ATPase, the cytosolic sensor Sestrin2 is required as well, even when the activating moiety is a lysosomally targeted amino acid methyl ester derivative. One possible explanation may be that amino acids are first imported into the lysosomes and then exported in a regulated manner (requiring intact lysosomes and lysosomal function) in order to be sensed by Sestrin2 and other such cytosolic sensors. The import step may depend on macropinocytosis (*Palm et al., 2015*; *Yoshida et al., 2015*) but the timing of the RAPTOR translocation step places a restraint on possible trafficking steps (see below). Our data cannot exclude the possibility that glutaminolysis in synergy with leucine also activate mTORC1 in the HAP-1 cells (*Durán et al., 2012*) but we consider it less likely to operate in HEK-293 cells where glutamine is absent from any stimulation medium.

The speed with which RAPTOR translocates to the lysosomes upon amino acid addition eliminates some potential routes of amino acid delivery to lysosomes, a question still unanswered. We consider it unlikely that import follows a canonical vesicular route, given the fact that fluid phase delivery to lysosomes takes longer than 10 min [*Araki (2006)* and our unpublished data] whereas RAPTOR translocation is evident within 2 min. This is also the case for macropinocytotic delivery to lysosomes (*Yoshida et al., 2015*). Alternatively, import into the cytosol via transporters (or via macropinosomes from where amino acids would be exported into the cytosol) would be fast enough, but then an equally fast second step would be needed for the import into the lysosomes. It will be important to understand the dynamics of this step, especially in view of the fast kinetics reported here. The data with the fluorescent analogue and with the di-leucine methyl ester argue that all of these steps can be bypassed completely as long as the activating moiety is delivered to functional lysosomes, in agreement with previous work (*Zoncu et al., 2011*). For both the methyl ester and the fluorescent analogue we have looked but did not see any macropinocytosis activation (our unpublished data).

Our work also revealed an apparent uncoupling between the translocation of mTORC1 to the lysosomes and the kinetics of mTORC1 target phosphorylation. In both HAP-1 cells edited to express endogenous RAPTOR-GFP and in HEK-293 cells stained for endogenous mTOR, phosphorylation of S6K and ULK1, major and direct mTORC1 targets, persisted long after translocation to the lysosomes had peaked and had returned to levels approaching the base line. This result implies that activated mTORC1 may exist in the cytosol, and not exclusively on the lysosomal surface. In line with this, a direct target of mTORC1 in this pathway, the protein 4EBP-1, was never detected to accumulate on lysosomes (*Figure 1*; *Figure 1—figure supplement 1*). We have also provided data showing that RHEB, an essential protein for mTORC1 activation, was not concentrated on lysosomes under any conditions in these cells, being partially localised to the ER/Golgi endomembrane system (*Figure 1*;

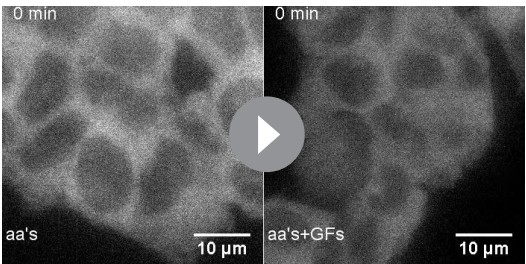

**Video 3.** Single imaging of RAPTOR-GFP HAP-1 cells: comparison of response with amino acids vs. amino acids and growth factors. After 55' of starvation in the incubator, cells were imaged for additional 5' of starvation on stage followed by on-stage addition of (1) amino acid containing starvation medium or (2) amino acid plus insulin plus EGF containing starvation medium. See also *Figure 5D and F*. Display is at 15 frames per second.

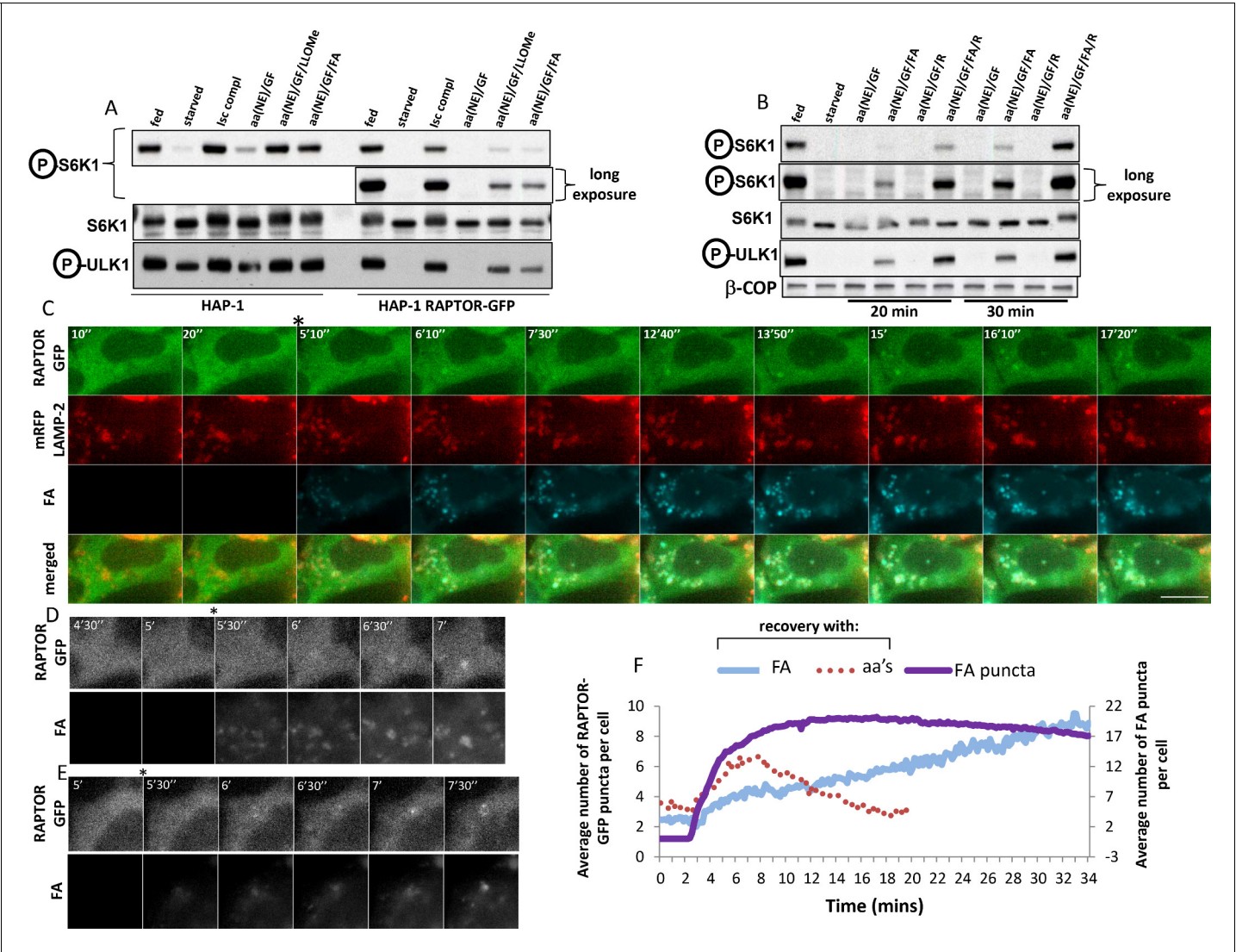

**Figure 6.** Dynamics of RAPTOR-GFP translocation to lysosomes containing a fluorescent amino acid analogue. (**A**) Cells – parental HAP-1 or those expressing RAPTOR-GFP – were kept in normal medium (Iscove's modified DMEM plus 10% FBS) or starved of amino acids and growth factors for 60' in medium containing salts and 1% dialysed FBS. The cells were then re-stimulated for 20' either with normal growth medium (Isc) or with a mixture of non-essential (NE) amino acids and growth factors (GF) added to the starvation medium and also containing in some samples 0.1 mM LLOMe or 0.4 mM fluorescent analogue (FA) as indicated. After lysis and electrophoresis the samples were immunoblotted for S6K phospho T389 and ULK1 phospho S757 as indicated. (**B**) HAP-1 cells expressing RAPTOR-GFP were starved and re-stimulated as indicated above, but with the addition in some cases of 0.4 mM arginine (R) as indicated. After lysis and electrophoresis the samples were immunoblotted for S6K phospho T389 and ULK1 phospho S757 as indicated. (**C–E**) HAP-1 cell expressing RAPTOR-GFP were transfected with plasmids encoding mRFP-LAMP2 for 24 hr and then moved to a microscope stage for live imaging experiments. After 5' in starvation medium, the cells were stimulated with medium containing NE amino acids, growth factors, arginine and the fluorescent analogue as described in (**B**) above. Images were captured every 10''. Selected frames are shown in the three channels and as an overlay for panel **C**. Bar represents 10 μm. Only the GFP and fluorescent analogue channels are shown for (**D–E**). (**F**) The number of puncta containing RAPTOR-GFP upon stimulation with amino acids and growth factors (aa's) or with the fluorescent leucine analogue (FA) were counted in each cell using the Spots Detection function of Imaris software (Bitplane/Andor), and the average of all cells in each condition is plotted here. The plot also contains the number of puncta containing the fluorescent analogue (FA puncta) during the stimulation. The response to the fluorescent analogue comes from a total of 125 cells measured from seven different coverslips, each with 3 different regions of interest. Although not shown here for clarity reasons, the standard deviation for each measurement is comparable to that shown in *Figure 5B and D*. The amino acid response shown here comes from 59 cells and only the early points are shown for clarity. Because of the smaller number of cells this part of the plot is more bumpy than the ones shown in *Figure 5* but the kinetics were remarkably similar.

The following source data and figure supplement are available for figure 6:

**Source data 1.** Raw data file (excel) for *Figure 6* plot F.

*Figure 6 continued on next page*

*Figure 6 continued*

**Figure supplement 1.** A leucine-based fluorescent analogue that can activate mTORC1.

*Figure 1—figure supplement 2*). Even when the Golgi localization of RHEB was disrupted with BFA (a treatment that had a very modest effect on mTORC1 activity), very little of the protein co-localized with RAGC, a major mTORC1 regulator residing on lysosomes (*Figure 1*; *Figure 1—figure supplement 3A and B*). All of this data in combination re-emphasise the importance of the lysosome as a key organelle involved in mTORC1 signalling but they provide some important modifications to the existing model. According to the current view lysosomes serve as platforms for the interaction between the mTORC1 complex (translocated there via its interaction with the RAGs and the Ragulator) and RHEB, leading to mTORC1 activation in situ. However, an alternative possibility is also worth considering in our view. It is possible that mTOR requires translocation to the lysosomal surface as part of the amino acid sensing function, but, subsequent to that, the complex comes off the membrane rapidly in order to phosphorylate its substrates. According to this model, the lysosomal surface may not be a terminal destination where activated mTOR resides but rather a platform that casts activated mTORC1 into the cytosol (*Figure 5*; *Figure 5—figure supplement 1C*). Having the activated complex free to move in the cytosol would be consistent with the fact that the known mTORC1 substrates, which number in the thousands, range from cytosolic to membrane-bound proteins, and to some with nuclear localization (*Hsu et al., 2011*). This idea would also be consistent with the fact that very little visible RHEB resides on the lysosomes under any condition, and that the TSC2 localization on and off lysosomes is cell-type dependent and not always able to explain the dynamics of mTORC1 inactivation on the basis of TSC2 lysosomal localization (*Demetriades et al 2014*; *Menon et al, 2014*; *Fawal et al, 2015*).

This dynamic view of the mTORC1 activation cycle requires answering the following question: what happens to mTORC1 on the lysosomal surface that allows activation to persist at later times? Two core proteins directly involved in mTORC1 activation are the RAG GTPases and RHEB, with the RAGs mediating mTORC1-RHEB interaction, and RHEB having an irreplaceable role as a co-factor during activity. In our edited cells, we were able for the first time to immunoprecipitate quantitatively all of the endogenous RAPTOR protein and to examine its binding partners. We consistently saw in RAPTOR/mTOR immunoprecipitates dynamic binding of the RAGA protein to the complex: it was greatly diminished during starvation and then re-bound to the complex early after amino acid recovery only to diminish again at the last time point (*Figure 5G*). Of note, when - based on live imaging - the RAPTOR-GFP translocation to the lysosomes was back to low levels during amino acid recovery, the protein was still bound to RAGA based on the immunoprecipitation experiments. This would suggest the possibility that translocation to the lysosomal surface may allow mTORC1 to acquire activated RAG proteins before returning to the cytosol. This sustained interaction with the RAG proteins away from the lysosomal surface may generate a "primed" mTORC1 complex which can interact with RHEB in various locations and lead to activity.

## Materials and methods

All chemicals were obtained from Sigma-Aldrich with the exception of concanamycin A which was from Tocris.

### Cell lines, antibodies, plasmids, siRNAs

HEK-293 cells were obtained as part of an adenovirus expression system from Microbix (https://microbix.com/what-we-do/adenovirus-vectors/

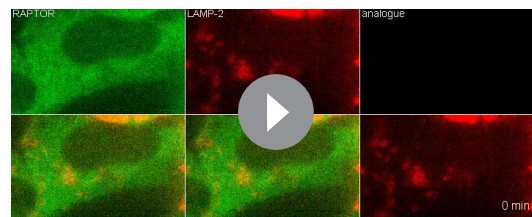

**Video 4.** Triple imaging of RAPTOR-GFP cells expressing mRFP-LAMP2 and incubated with fluorescent analogue. Starved cells were imaged for 5' as above and fluorescent analogue was then added on stage with continuous imaging. Different overlays are shown in addition to the single channels. See also *Figure 7C*. Display is at 5 frames per second.

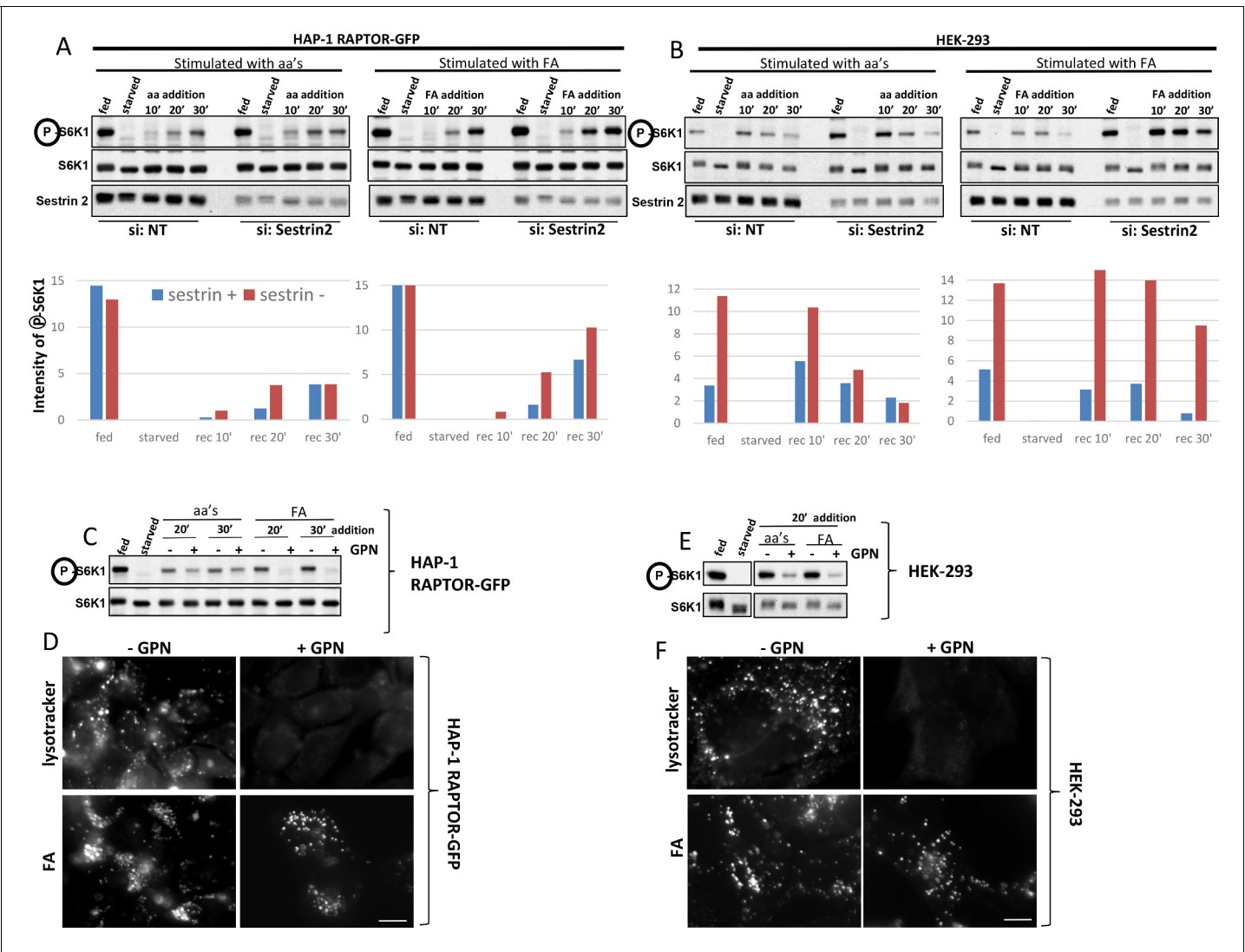

**Figure 7.** Activation of mTORC1 in HAP-1 RAPTOR-GFP cells and in HEK-293 cells depends on Sestrin2 and intact lysosomes. (**A**) HAP-1 cells expressing RAPTOR-GFP were treated with siRNA against Sestrin2 or with a non-targetting control (NT) as indicated. After 72 hr, cells were kept in normal medium (Iscove's modified Dulbecco's medium -IMDM plus 10% FBS) or starved of amino acids and growth factors for 60' in medium containing salts and 1% dialysed FBS. The starved cells were then re-stimulated for 10', 20' or 30' with a mixture of amino acids and growth factors (aa) or with fluorescent analogue (fa) added to the starvation medium as indicated. After lysis and electrophoresis the samples were immunoblotted for the indicated proteins. The intensity of the S6K phospho T389 band is plotted in the graphs. This experiment is representative of two. (**B**) The same experiment as in A was done with HEK-293 cells. This experiment is representative of two. (**C**) HAP-1 cells expressing RAPTOR-GFP were kept in normal medium or starved and re-stimulated for 20' and 30' with amino acids and growth factors (aa) or with fluorescent analogue (FA) in the presence or absence of GPN as indicated. After lysis and electrophoresis the samples were immunoblotted for the indicated proteins. (**D**) Cells as in C were loaded with lysotracker or with fluorescent analogue (FA) for 30' during aa re-stimulation in the presence or absence of GPN as indicated. After fixation the cells were examined by fluorescence microscopy. (**E**) HEK-293 cells were kept in normal medium or starved and re-stimulated for 20' with MEM amino acids (aa) or with fluorescent analogue (FA) in the presence or absence of GPN as indicated. After lysis and electrophoresis the samples were immunoblotted for the indicated proteins. (**F**) Cells as in E were loaded with lysotracker or with fluorescent analogue (FA) for 20' during aa re-stimulation in the presence or absence of GPN as indicated. After fixation the cells were examined by fluorescence microscopy. The bar in all panels represents 10 μm.

The following figure supplement is available for figure 7:

**Figure supplement 1.** Properties of the fluorescent leucine analogue in mTORC1 activation.

adenovirus-price-list/) and were tested for mycoplasma contamination every 2 years in our lab using Mycoguard (Genecopoeia, http://www.genecopoeia.com/product/mycoplasma-detection-kit/). HAP-1 cells and the RAPTOR-GFP edited cells were obtained from Haplogen (now part of Horizon Genomics https://www.horizondiscovery.com/) and were tested for mycoplasma contamination every 2 years. The following antibodies were used. Mouse anti-β-COP (a kind gift from the late Thomas Kreis), mouse anti-LAMP2 (Developmental Studies Hybridoma Bank, PRID: AB_528129), rabbit anti-phosphoS6K1 (Cell Signaling, PRID: AB_330944), rabbit anti-phosphoS6 (Cell Signaling, RRID: AB_916156), rabbit anti-mTOR (Cell Signaling, RRID:AB_2105622), rabbit anti-RAGC (Cell Signaling, PRID:AB_2180068), rabbit anti-RAPTOR (Cell Signaling, PRID:AB_10694695), mouse ant-RHEB (Abnova, PRID:AB_1112097), rabbit anti-phospho4EBP1 (Cell Signaling, PRID: AB_560835), rabbit anti-Giantin (Covance, PRID:AB_291560), rabbit anti-phosphoULK1 (Cell Signaling PRID: AB_10829226), rabbit Sestrin2 (Proteintech Group, PRID:AB_2185480). The following Plasmids were used. pRK5 HA GST Rheb1 from D Sabatini (Addgene 14951), RAGB from D Sabatini (Addgene 19301), myc-RAPTOR from D Sabatini (Addgene 1859). RAPTOR-EGFP was generated from myc-RAPTOR by excising the RAPTOR open reading frame, tailing it with restriction sites for Xho1 and EcoR1 and cloning it into the pEGFP-C1 vector. To target endogenous RAPTOR we used the following oligos:

si-1 targetting 3′ untranslated region with sequence A.G.A.G.A.G.A.G.G.A.A.G.A.A.G.G.A.G.A. U.U

si-2 targeting 5′ untranslated region with sequence G.G.G.C.U.G.A.U.G.A.G.A.U.G.A.G.U.U.U.U. U.

## Generation of RAPTOR-GFP cell line

The RAPTOR-GFP cell line was created using a strategy based on that of Auer, et al (*Auer et al., 2014*). HAP-1 cells were transfected with several plasmids: (1) a plasmid expressing Cas9, (2) a donor plasmid containing the GFP coding sequence flanked by zebra fish-specific guide RNAs and the zebra fish guide RNA sequence under the U6 promoter, (3) a plasmid expressing a guide RNA sequence (TGGAGAAGCGTGTCAGATAG) targeting the 3′ end of the RAPTOR gene. When the donor plasmid is cleaved in the transfected cells, it will likely be integrated in the site targeted by the RAPTOR guide RNA. GFP positive transfected cells were sorted using FACS and subjected to single cell dilution to obtain clonal cell lines. These clonal cell lines were screened to search for cells which contained the GFP cassette integrated at 3′ end of the RAPTOR gene. These cells are commercially available from Horizon Genomics.

## Generation of fluorescent amino acid analogue

The synthetic route outlined in the scheme shown in Supplementary *Figure 4B* was followed using standard methods. All reagents are commercially available. The final product was freeze dried from water plus 5% acetic acid and re-suspended at 0.5 M in DMSO as a stock solution kept at –20°C.

## Starvation of cells and re-stimulation with aa's

For starvation (both HEK-293 and HAP-1), cells were washed twice with pre-warmed starvation medium (140 mM NaCl, 1 mM $CaCl_2$, 1 mM $MgCl_2$, 5 mM Glucose, 20 mM Hepes, 5 mM KCl, pH 7.4) containing either 1% BSA or 1% dialysed FBS prior to incubation with this medium for 60 min. At the end of 60 min, the medium for HEK-293 cells was replaced with new starvation medium containing 2X solution of MEM amino acid solution (50X stock, containing the following amino acids in one letter code R, C, H, I, L, K, M, F, T, W, Y, V, and sold by LifeTechnologies), and incubated in this solution for the indicated times. For HAP-1 cells, the starvation medium was replaced with new starvation medium containing 1 mM glutamine, MEM amino acids as above, 2X solution of NE amino acids (100X stock, containing the following amino acids in one letter code G, A, N, D, E, P, S and sold by LifeTechnologies), insulin (1:500 dilution of liquid supplement at 9–11 mg/ml sold by Sigma-Aldrich) and EGF (20 ng/ml) as indicated in the Figure legends. In some experiments using the fluorescent analogue, we also added 0.4 mM arginine to the above solution.

## Immunofluorescence and live imaging

Cells for immunofluorescence were grown on glass coverslips and fixed in 3.7% formaldehyde in 200 mM Hepes pH 7.2. Staining for immunofluorescence and digital photography were done as described before (*Karanasios et al., 2013*). Live-cell imaging was performed as previously described (*Karanasios et al., 2013*) using a Nikon Ti-E-based system. Cells plated onto 60 plastic dishes were transferred onto 22-mm-diameter glass coverslips (BDH) and secured in an imaging chamber with 2 ml of cell medium or starvation medium added as indicated. The assembled imaging chamber was secured onto the microscope stage, and cells were maintained at 37°C using an OKO Labs full enclosure incubation system. The Nikon Ti-E-based system comprised a Nikon Ti-E microscope, 100x 1.4 N.A. objective (Nikon), SpecraX LED illuminator (Lumencor, Beaverton, OR), 410/504/582/669-Di01 and Di01-R442/514/561 dichroic mirrors (Semrock), Hamamatsu Flash 4.0 sCMOS camera, emission filter wheel (Sutter Instruments) and was controlled using Nikon Elements software. Compounds (fluorescent analogue, amino acids, and drugs) were added during imaging by flushing the solution in the imaging chamber with 5 ml of fresh solution containing the indicated additions.

## Acknowledgements

This work is supported by the Biotechnology and Biological Sciences Research Council [grant number BB/K019155/1 to NTK]. We thank our colleagues Len Stephens, Phill Hawkins and Roger Williams for many useful discussions.

## Additional information

### Funding

| Funder | Grant reference number | Author |
|---|---|---|
| Biotechnology and Biological Sciences Research Council | core funding | Maria Manifava<br>Matthew Smith<br>Sergio Rotondo<br>Simon Walker<br>Izabella Niewczas<br>Roberto Zoncu<br>Jonathan Clark<br>Nicholas T Ktistakis |
| Biotechnology and Biological Sciences Research Council | BB/K019155/1 | Nicholas T Ktistakis |

The funders had no role in study design, data collection and interpretation, or the decision to submit the work for publication.

### Author contributions

MM, SR, SW, Conception and design, Acquisition of data, Analysis and interpretation of data; MS, NTK, Conception and design, Acquisition of data, Analysis and interpretation of data, Drafting or revising the article; IN, Acquisition of data, Analysis and interpretation of data; RZ, Conception and design, Analysis and interpretation of data, Contributed unpublished essential data or reagents; JC, Conception and design, Analysis and interpretation of data

### Author ORCIDs

Simon Walker, http://orcid.org/0000-0001-9185-4922
Nicholas T Ktistakis, http://orcid.org/0000-0001-9397-2914

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
