## [Decision Letter]

[Editors’ note: a previous version of this study was rejected after peer review, but the authors submitted for reconsideration. The first decision letter after peer review is shown below.]

Thank you for submitting your work entitled "Dynamics of mTORC1 activation in response to amino acids" for consideration by *eLife*. Your article has been reviewed by two peer reviewers, and the evaluation has been overseen by a Reviewing Editor and Tony Hunter as the Senior Editor. Our decision has been reached after consultation between the reviewers. Based on these discussions and the individual reviews below, we regret to inform you that your work will not be considered further for publication in *eLife*.

The reviewers found your work of interest but were very critical about the lack of controls and incorrect interpretation of the data. If you believe that you could improve the work, we would be happy to consider the work as a new submission. However, given the extensive requirements specified by the reviewers, we would understand if you choose to submit this work elsewhere.

1) Being able to follow Raptor translocation with endogenous levels of protein could be valuable, but it appears that the Raptor-GFP fusion protein is a hypomorph, and does not fully recapitulate the function of native Raptor. For example, amino acids alone failed to activate mTORC1 in the Raptor-GFP HAP-1 cells unlike the parental HAP-1 cells, and maximal activation of mTORC1 in Raptor-GFP HAP-1 cells required both amino acids and growth factor stimulation, and even then it was not as strong as in parental HAP-1 cells. In this regard, both growth factors and amino acids are generally required for the full activation of mTORC1 in in mammalian cells, and therefore it is not clear why mTORC1 was fully activated in HAP-1 cells just by replenishing amino acids.

2) The lag in S6K phosphorylation behind Raptor recruitment to lysosomes looks real, but the authors need to demonstrate that the anti-pT389 S6K blotting signal is linear before drawing a firm conclusion (it would be ideal if they could somehow monitor S6K phosphorylation in single cells in parallel with Raptor-GFP translocation).

3) In contrast to the authors' conclusion, the immunofluorescence images in Figure 1 show overlap between LAMP-1 and mTOR at all times. The authors should repeat the experiments with a Raptor antibody.

4) Several figures: When assessing S6K, 4E-BP1 and S6 phosphorylation by immunoblotting, the authors should also blot for total levels of these proteins.

5) Figure 1, Figure 2; Figure 1—figure supplement 1; Figure 1—figure supplement 3; Figure 5—figure supplement 5A-B; and Figure 6—figure supplement 1. The authors should include a loading control.

6) Figure 2 – the authors should determine if RAPTOR-GFP is localized to the lysosomes. In Figure 3 the authors showed some co-localization with LAMP-1. However, this was not quantified and most of the GFP-RAPTOR is not co-localized with LAMP-1. In addition there was not much change in GFP-RAPTOR localization in starved vs. fed conditions (Figure 3).

To determine the functionality of RAPTOR-GFP, the authors exclusively rely on S6K phosphorylation measurements by immunoblot. It would be informative to measure other readouts such as 4E-BP1.

7) Figure 1—figure supplement 2. The authors should blot for total levels S6K (or HA), RHEB and RAGA in the lysate and immunoprecipitated samples. As is, they check only the phosphorylated form of S6K. Also, a negative control for the immunoprecipitation is missing.

8) The authors did not provide mechanistic explanation for the observations presented in Figure 5 and Figure 6. Based on experiments using a fluorescent leucine methyl ester analogue, the authors seem to favor the "inside out" model of amino acid sensing in which amino acids are sensed in the lysosomal lumen. However, a recent study showed that cytoplasmic SESN2 is "almost certainly" the major leucine sensor upstream of mTORC1 (Wolfson RL et al. Science. 351:43). This recent study should be mentioned and discussed. How do the authors know that the mechanism of sensing the leucine analogue is the same as that of sensing a non-esterified leucine (based on the structure of the leucine-binding pocket in SESN2, it seems unlikely to accommodate this analogue)? The authors should determine if SESN2 depletion affects translocation of RAPTOR-GFP to lysosomes upon addition of the leucine analogue. If it does not, how do the authors reconcile their findings with those of Wolfson et al.?

Alternatively, if SESN2 depletion has an effect, how can the authors argue that accumulation in the lysosome lumen is physiologically relevant? The glutaminolysis model, which provides a sensing mechanism for leucine and glutamine in the mitochondria, should also be mentioned (Durán RV et al. Mol Cell. 2012. 47:349).

[Editors’ note: what now follows is the decision letter after the authors submitted for further consideration.]

Thank you for choosing to send your work entitled "Dynamics of mTORC1 activation in response to amino acids" for consideration at eLife. Your resubmission has been evaluated by Tony Hunter and the original Reviewing Editor who handled your submission. The editors are generally satisfied with the revisions made, but would like to ask you to address the following points.

On reading the resubmitted paper in depth, the conclusion drawn from the new data in Figure 7 is somewhat puzzling. In these experiments the authors have addressed the requirement for Sestrin2 (Sesn2), a recently reported Leu sensor, for mTORC1 stimulation by their new fluorescent Leu analogue. Based on the work from the Sabatini lab and others, the prevailing current model is that Sesn2 binds to and inhibits the GATOR2 complex, which in turn inhibits GATOR1, a complex that acts as a RagA/RagC GAP to prevent Ragulator recruitment of mTORC1 to the lysosome. Leu binding to Sesn2 causes it to dissociate from GATOR2 allowing to inhibit GATOR1, which then leads to RagA/RagC activation and mTORC1 recruitment to the lysosome where it interacts with Rheb, i.e. GATOR2 is effectively an activator of mTORC1, and not "a key negative regulator of mTORC1", as the authors state in subsection “Activation of mTORC1 by the amino acid analogue depends on Sestrin2 and on intact lysosomes”. Based on this model, the knockdown of Sesn2 and the consequent loss of its GATOR2 inhibitory activity would be expected to result in (partial) activation of mTORC1 in the absence of Leu (which is what the authors observed in HEK 293 cells). Moreover, if Sens2 were the only Leu sensor in the system, then one would expect mTORC1 activation to be insensitive to Leu in cells devoid of Sesn2.

1) The new data in Figure 7 show that partial depletion of Sesn2 increased mTORC1 stimulation by the FA Leu analogue and by amino acids, which on the face of it does not fit with the model described above. It is possible that Sesn2 is in excess over GATOR2, and that, even though Sesn2 levels have been reduced by si-Sestrin2, the remaining level of Sesn2 protein is sufficient to bind and inhibit all the GATOR2 complex, and that the lower level of Sesn2 somehow sensitizes the system to Leu (it would be better to use CRISPR technology to knock out Sesn2, but this would be beyond the scope of the paper). Other possible explanations are that Sesn1 is partially redundant with Sesn2 for Leu sensing in these cells or there is a second Leu sensor in the system that is affected by Sesn2 in some way. Ideally, what the authors should do is to re-express siRNA-resistant WT Sesn2 (this is a control for potential off target effects of si-Sestrin2 that is missing from Figure 7) and also Sesn2 L261A, which cannot bind Leu (did the authors try to model the FA Leu analogue into the Leu-binding pocket of Sesn2?) in the si-Sestrin2 treated cells, and check the response to FA Leu and amino acids.

2) It would strengthen these data if GFP-Raptor translocation to lysosomes FA Leu and amino acid-induced were monitored in the Sesn2 knock HAP1-RAPTOR-GFP cells.

In sum, please provide a response to these issues and explain the apparent paradox that cells remain responsive to Leu after removal of a Leu sensor from the system.

---

## [Author Response]

[Editors’ note: the author responses to the first round of peer review follow.]

*The reviewers found your work of interest but were very critical about the lack of controls and incorrect interpretation of the data. If you believe that you could improve the work, we would be happy to consider the work as a new submission. However, given the extensive requirements specified by the reviewers, we would understand if you choose to submit this work elsewhere.*

We thank the reviewers for the thorough evaluation of the manuscript and for suggesting important experiments to follow up. Below are our responses to specific comments.

One comment we want to make that may not have been clear in the original manuscript. The dynamic range of the fluorescence signal of the RAPTOR-GFP edited cells is much lower than a typical cell overexpressing a GFP-tagged protein, owing to the fact that expression in HAP-1 cells rely on the endogenous promoter. This limits to some extent the types of quantitations that we can do since small differences are unlikely to be reproducibly measured. The measurements that we reported are done with a lot of confidence but we cannot push the system further. To make this point clear I am attaching photos from the RAPTOR-GFP cells and HEK-293 cells expressing GFP-ATG13 together with the dynamic range of their intensity. Please note that the range for RAPTOR-GFP is 102-250 whereas for GFP-ATG13 is 5 times larger at 379-1088.

Author response image 1.**DOI:**
http://dx.doi.org/10.7554/eLife.19960.022

*1) Being able to follow Raptor translocation with endogenous levels of protein could be valuable, but it appears that the Raptor-GFP fusion protein is a hypomorph, and does not fully recapitulate the function of native Raptor. For example, amino acids alone failed to activate mTORC1 in the Raptor-GFP HAP-1 cells unlike the parental HAP-1 cells, and maximal activation of mTORC1 in Raptor-GFP HAP-1 cells required both amino acids and growth factor stimulation, and even then it was not as strong as in parental HAP-1 cells. In this regard, both growth factors and amino acids are generally required for the full activation of mTORC1 in in mammalian cells, and therefore it is not clear why mTORC1 was fully activated in HAP-1 cells just by replenishing amino acids.*

There are two issues here.

A) The edited cells being hypomorphs. This is correct and we have carefully documented it. However, we still were able to set up these cells to depend critically on amino acids for activation, and to re-capitulate the translocation of the mTORC1 component RAPTOR. Wherever possible, we have also supplemented the data with experiments from HEK-293 and parental HAP-1 cells. The original Sancak et al. paper describing for the first time mTOR translocation to lysosomes was published in 2008, but no live imaging for this phenomenon has been published since then to our knowledge. This is probably indicative of the problems in setting up this experiment. We think that the fact that the edited cells are hypomorphic must be evaluated against the fact that they are the first reagent to allow live imaging of this phenomenon.

B) Interplay between amino acids and grown factors in mTORC1 activation. Here we have included a new experiment (Figure 2) to show effects of serum and/or amino acid starvation in subsequent mTORC1 activation by amino acids and growth factors in HEK-293 or HAP-1 cells. This experiment shows serum starvation minimises mTORC1 activity which can then be stimulated with growth factors in HEK-293, whereas HAP-1 cells are resistant to this. However, both cells are sensitive to amino acid starvation and re-stimulation. Because in this work we are interested in the speed and dynamics of re-stimulation we chose from the beginning to use a 60 min total (amino acid and serum) starvation.

*2) The lag in S6K phosphorylation behind Raptor recruitment to lysosomes looks real, but the authors need to demonstrate that the anti-pT389 S6K blotting signal is linear before drawing a firm conclusion (it would be ideal if they could somehow monitor S6K phosphorylation in single cells in parallel with Raptor-GFP translocation).*

We have addressed this point by including a dose response blot for phospho-S6K in Figure 5 (bottom two blots) and by adding to most experiments in the paper blots for phospho-ULK1, another direct target of mTORC1. These additional data strengthened the original conclusion.

The last comment is of course the ideal experiment. It would require transfecting to the edited cells a FRET reporter for S6K activation and then looking at activity and translocation on a cell-by-cell basis. Several years ago a paper reported such a construct (Komatsu et al., (2011) MBC 22: 4647-56) and we obtained it at the time from the lab that published it with a view to use is as described above. We tried at the time to see if the reporter would show FRET after a round of amino acid starvation and re-stimulation, or after overnight serum starvation and re-stimulation with growth factors but it did not. We think that this reporter was oversold and that such a FRET reporter for S6K does not exist at present.

*3) In contrast to the authors' conclusion, the immunofluorescence images in Figure 1 show overlap between LAMP-1 and mTOR at all times. The authors should repeat the experiments with a Raptor antibody.*

We clarify what the experiment in Figure 1 measures: we measured total intensity of staining per cell for LAMP-1 and mTOR for the same cells. Threshold was kept constant for *all* mTOR and *all* LAMP-1, but different between them to optimize the accuracy of the measurement. What we are claiming here is that whereas the intensity of LAMP-1 is constant, the intensity of mTOR peaks and then comes down. It is true that whatever pool of mTOR is still left in the intense staining region co-localizes with LAMP-1: what we are saying is that this amount/pool of mTOR is not constant but it peaks and comes down. For this submission we have repeated this experiment with somebody else’s hands and we get the same result.

In order to provide an alternative quantitative description of the translocation of RAPTOR to lysosomes we derived a Pearson’s coefficient from live cells expressing RAPTOR-GFP and LAMP2; this is shown in new Figure 4.

In order to satisfy the request of examining this phenomenon with RAPTOR staining, we ordered and tried 5 different RAPTOR antibodies (Cell Signaling, Proteintech, Millipore). None works for iffl as shown by our siRNA experiments and we were unable to find in the literature reference to an antibody for RAPTOR validated for iffl.

*4) Several figures: When assessing S6K, 4E-BP1 and S6 phosphorylation by immunoblotting, the authors should also blot for total levels of these proteins.*

We have added the requested blots for the non-phosphorylated versions of these proteins.

*5) Figure 1, Figure 2, Figure 1—figure supplement 1, Figure 1—figure supplement 3, Figure 5—figure supplement 5A-B and Figure 6—figure supplement 1. The authors should include a loading control.*

Wherever missing we have added a loading control. Please note that we tend to rely on the non-specific upper band revealed by the phospho-S6K antibody (marked with an asterisk in some blots), which, in our experience, is as good indicator of loading as any other band. We also load equivalent amount of protein in our blots after protein quantitation with the BCA reagent.

*6) Figure 2 – the authors should determine if RAPTOR-GFP is localized to the lysosomes. In Figure 3 the authors showed some co-localization with LAMP-1. However, this was not quantified and most of the GFP-RAPTOR is not co-localized with LAMP-1. In addition there was not much change in GFP-RAPTOR localization in starved vs. fed conditions (Figure 3).*

To address this question we quantitated the extent of RAPTOR/LAMP2 co-localization using the Pearson’s coefficient in live cells and in fixed cells. The plot for live cells is in Figure 4 and the values in fixed cells is in Figure 4. In both cases, amino acid stimulation increases the co-localization between RAPTOR and LAMP2 by 2-3 fold. For the live cell experiment we included a Pearson’s channel showing the co-localizing puncta in the revised Video 1. We also tried to quantitate the percentage of total RAPTOR that moves from the cytosol to the lysosomal locus during this stimulation, but are not yet comfortable with the data and therefore would rather not show them.

*To determine the functionality of RAPTOR-GFP, the authors exclusively rely on S6K phosphorylation measurements by immunoblot. It would be informative to measure other readouts such as 4E-BP1.*

For most blots, and certainly for all the critical ones, we have re-blotted for phospho-ULK1 (Ser 757), which is a very reliable reporter of mTORC1 activity directly linked to autophagy. These new data re-enforce our conclusions derived from phospho-S6K blots.

*7) Figure 1—figure supplement 2. The authors should blot for total levels S6K (or HA), RHEB and RAGA in the lysate and immunoprecipitated samples. As is, they check only the phosphorylated form of S6K. Also, a negative control for the immunoprecipitation is missing.*

We have repeated this transfection and the new data are shown in Figure 1—figure supplement 2. Our data are consistent with what we reported before, and please note that this strategy for activating mTOR has been used by us in other publications as well (Musiwaro et al., Autophagy 2013) so this is a well-controlled assay in the lab. In the process of re-doing this figure we realised that our control transfection was with RAGB and not RAGA. We apologise for this mistake in the previous version.

*8) The authors did not provide mechanistic explanation for the observations presented in Figure 5 and Figure 6. Based on experiments using a fluorescent leucine methyl ester analogue, the authors seem to favor the "inside out" model of amino acid sensing in which amino acids are sensed in the lysosomal lumen. However, a recent study showed that cytoplasmic SESN2 is "almost certainly" the major leucine sensor upstream of mTORC1 (Wolfson RL et al. Science. 351:43). This recent study should be mentioned and discussed. How do the authors know that the mechanism of sensing the leucine analogue is the same as that of sensing a non-esterified leucine (based on the structure of the leucine-binding pocket in SESN2, it seems unlikely to accommodate this analogue)? The authors should determine if SESN2 depletion affects translocation of RAPTOR-GFP to lysosomes upon addition of the leucine analogue. If it does not, how do the authors reconcile their findings with those of Wolfson et al.?*

*Alternatively, if SESN2 depletion has an effect, how can the authors argue that accumulation in the lysosome lumen is physiologically relevant? The glutaminolysis model, which provides a sensing mechanism for leucine and glutamine in the mitochondria, should also be mentioned (Durán RV et al. Mol Cell. 2012. 47:349).*

We have spent the majority of time trying to address this question and the results are shown in two new Figures (Figure 7 and Figure 7—figure supplement 1). First we determined whether down-regulation of Sestrin2 by siRNA would enhance activation via amino acids and/or the fluorescent analogue, and this data is in Figure 7 and B. For both HEK-293 and edited HAP-1 cells we found that Sestrin2 is a negative regulator of the response even when stimulation was with the fluorescent analogue-in fact the effects with the fluorescent analogue are stronger. We also determined that amino acids and the fluorescent analogue require intact lysosomes to activate mTORC1 in both HAP-1 and HEK-293 cells (Figure 7). This is the case although there is fluorescent analogue associating with the lysosomes even when they are ruptured with GPN. Finally, we determined that whereas amino acid re-stimulation of mTORC1 is sensitive to the macropinocytosis inhibitor EIPA (this was done based on recent papers suggesting that macropinocytsis may underpin amino acid delivery), re-stimulation by di-leucine methyl ester or the fluorescent analogue is not (Figure 7—figure supplement 1). In our opinion all of this data argue that the methyl ester and the fluorescent analogue by-pass whatever macropinocytosis-related step is required for amino acid delivery and localise directly and quickly in the lysosomes. However, given the sensitivity of their activity to (a) vacuolar ATPase inhibitors and (b) intact lysosomes we believe that they then require exit from the lysosome in a form that can be sensed by sestrin2 in order to activate mTORC1.

[Editors' note: the author responses to the re-review follow.]

*On reading the resubmitted paper in depth, the conclusion drawn from the new data in Figure 7 is somewhat puzzling. In these experiments the authors have addressed the requirement for Sestrin2 (Sesn2), a recently reported Leu sensor, for mTORC1 stimulation by their new fluorescent Leu analogue. Based on the work from the Sabatini lab and others, the prevailing current model is that Sesn2 binds to and inhibits the GATOR2 complex, which in turn inhibits GATOR1, a complex that acts as a RagA/RagC GAP to prevent Ragulator recruitment of mTORC1 to the lysosome. Leu binding to Sesn2 causes it to dissociate from GATOR2 allowing to inhibit GATOR1, which then leads to RagA/RagC activation and mTORC1 recruitment to the lysosome where it interacts with Rheb, i.e. GATOR2 is effectively an activator of mTORC1, and not "a key negative regulator of mTORC1", as the authors state in subsection “Activation of mTORC1 by the amino acid analogue depends on Sestrin2 and on intact lysosomes”. Based on this model, the knockdown of Sesn2 and the consequent loss of its GATOR2 inhibitory activity would be expected to result in (partial) activation of mTORC1 in the absence of Leu (which is what the authors observed in HEK 293 cells). Moreover, if Sens2 were the only Leu sensor in the system, then one would expect mTORC1 activation to be insensitive to Leu in cells devoid of Sesn2.*

We appreciate greatly the comments made by Tony Hunter and the original Reviewing Editor and the seriousness of their thinking. We have corrected the mistake of referring to GATOR2 “a key negative regulator of mTORC1” instead of a “key negative regulator of GATOR1” which is correct. The rest of the sentence and the concept described are correct.

*1) The new data in Figure 7 show that partial depletion of Sesn2 increased mTORC1 stimulation by the FA Leu analogue and by amino acids, which on the face of it does not fit with the model described above. It is possible that Sesn2 is in excess over GATOR2, and that, even though Sesn2 levels have been reduced by si-Sestrin2, the remaining level of Sesn2 protein is sufficient to bind and inhibit all the GATOR2 complex, and that the lower level of Sesn2 somehow sensitizes the system to Leu (it would be better to use CRISPR technology to knock out Sesn2, but this would be beyond the scope of the paper). Other possible explanations are that Sesn1 is partially redundant with Sesn2 for Leu sensing in these cells or there is a second Leu sensor in the system that is affected by Sesn2 in some way. Ideally, what the authors should do is to re-express siRNA-resistant WT Sesn2 (this is a control for potential off target effects of si-Sestrin2 that is missing from Figure 7) and also Sesn2 L261A, which cannot bind Leu (did the authors try to model the FA Leu analogue into the Leu-binding pocket of Sesn2?) in the si-Sestrin2 treated cells, and check the response to FA Leu and amino acids.*

With respect to question and comment 1, we think that the most likely explanation may be that Sestrin2 is in excess over GATOR2 and down-regulating it has an enhancing effect upon stimulation. Similar observations have been reported in two recent papers of which the relevant sections are noted below and which we have now added in the reference and discussed briefly in the text in subsection “Activation of mTORC1 by the amino acid analogue depends on Sestrin2 and on intact lysosomes”. [Our aim with the Sestrin2 data is not to resolve the debate as to its exact role but to provide evidence that both the edited cells and the fluorescent analogue work within the canonical pathway.]

(a) “Sestrin2 silencing in RKO cells enhanced the mTORC1 signaling in both starved and serumstimulated cells (Figure 7), indicating that endogenous Sestrin2 indeed functions to inhibit mTORC1 signaling in human colon cancer cells.” http://dx.doi.org/10.7554/eLife.12204.001

(b) “The sensitivity of mTORC1 signaling to amino acids was dependent on Sestrin gene dosage: the fewer alleles of Sestrins, the less dependence of MEFs on amino acids” Note in Figure 7 that in triple Sestrin KO cells amino acids enhance mTORC1 activity above the basal levels induced by FBS alone. http://dx.doi.org/10.1016/j.cell.2014.08.038

*2) It would strengthen these data if GFP-Raptor translocation to lysosomes FA Leu and amino acid-induced were monitored in the Sesn2 knock HAP1-RAPTOR-GFP cells.*

*In sum, please provide a response to these issues and explain the apparent paradox that cells remain responsive to Leu after removal of a Leu sensor from the system.*

With respect to question and comment 2, we had tried to see if in the Sestrin2 down-regulated cells we can observe a change in the dynamics of the RAPTOR translocation by live imaging. This was essentially what we tried to do before by modulating the levels of growth factors in the continuous presence of amino acids (Figure 5) since we had seen effects in phospho-S6K phosphorylation. Unfortunately our resolution of the response by live imaging is not high enough to see such differences (this was also the case when we looked at growth factors before), so we prefer to show only the blots (which we have done in two different cell lines).